



# RTTOV-gb - Adapting the fast radiative transfer model
# RTTOV for the assimilation of ground-based microwave
# radiometer observations
Francesco De Angelis[1], Domenico Cimini[2,1], James Hocking[3], Pauline Martinet[4], Stefan
Kneifel[5].
[1]CETEMPS, University of L'Aquila, L'Aquila, Italy
[2]IMAA-CNR, Potenza, Italy
[3]MET OFFICE, Exeter, United Kingdom
[4]METEO FRANCE – CNRM/GAME, Toulouse, France
[5]INSTITUTE FOR GEOPHYSICS AND METEOROLOGY, University of Cologne, Cologne, Germany
*Correspondence to:* F. De Angelis (francesco.deangelis1@graduate.univaq.it)
**Abstract.** Ground-based microwave radiometers (MWR) offer a new capability to provide continuous
observations of the atmospheric thermodynamic state in the planetary boundary layer. Thus, they are potential
candidates to supplement radiosonde network and satellite data to improve numerical weather prediction (NWP)
models through a variational assimilation of their data. However in order to assimilate MWR observations a fast
radiative transfer model is required and such a model is not currently available. This is necessary for going from
the model state vector space to the observation space at every observation point. The fast radiative transfer
model RTTOV is well accepted in the NWP community, though it was developed to simulate satellite
observations only. In this work, the RTTOV code has been modified to allow for simulations of ground-based
upward looking microwave sensors. In addition, the Tangent Linear, Adjoint, and K-modules of RTTOV have
been adapted to provide Jacobians (i.e. the sensitivity of observations to the atmospheric thermodynamical state)
for ground-based geometry. These modules are necessary for the fast minimization of the cost function in a
variational assimilation scheme. The proposed ground-based version of RTTOV, called RTTOV-gb, has been
validated against accurate and less time-efficient line-by-line radiative transfer models. In the frequency range
commonly used for temperature and humidity profiling (22-60 GHz), root-mean-square brightness temperature
differences are smaller than typical MWR uncertainties (~0.5 K) at all channels used in this analysis. Brightness
temperatures (TB) computed with RTTOV-gb from radiosonde profiles have been compared with nearly
simultaneous and colocated ground-based MWR observations. Differences between simulated and measured TB
are below 0.5 K for all channels except for the water vapor band, where most of the uncertainty comes from
instrumental errors. The Jacobians calculated with the K-module of RTTOV-gb have been compared with those
calculated with the brute force technique and those from the line-by-line model ARTS. Jacobians are found to be
almost identical, except for liquid water content Jacobians for which a 10% difference between ARTS and
RTTOV-gb at transparent channels around 450hPa is attributed to differences in liquid water absorption models.
Finally, RTTOV-gb has been applied as the forward model operator within a 1-Dimensional Variational (1D-
Var) software tool in an Observing-System Simulation Experiment (OSSE). For both temperature and humidity
profiles, the 1D-Var with RTTOV-gb improves the retrievals with respect to NWP model in the first few
kilometers from the ground.





## 1    Introduction

The planetary boundary layer (PBL) is the single most important under-sampled part of the atmosphere (National Research Council, 2008). While the thermodynamical state of the atmosphere is well measured at the surface by ground in-situ sensors and in the upper troposphere by satellite sounders, there is currently an observational gap in the PBL. According to the WMO Statement Of Guidance For Global Numerical Weather Prediction (WMO, 2014), there are four priorities for atmospheric variables not adequately measured in the PBL: wind profiles, temperature and humidity profiles (in cloudy areas), precipitation, and snow mass. Ground-based microwave radiometers (MWR) provide temperature and humidity profiles in both clear- and cloudy-sky conditions with high temporal resolution and low-to-moderate vertical resolution, with information mostly residing in the PBL (Cimini et al., 2006). Ground-based MWR offer to bridge the current observational gap by providing continuous temperature and humidity profiles in the PBL. When combined with satellite observations, the total information content of the derived atmospheric profiles can be significantly enhanced (Ebell et al., 2013). The data assimilation (DA) of MWR observations into numerical weather prediction (NWP) models may be particularly important in nowcasting and severe weather (fog, convection, turbulence, etc.) initiation. The assimilation of MWR data has been recently investigated (Cimini et al., 2014; Caumont et al 2015), assimilating temperature and humidity profile retrievals from a network of 13 MWR members from the international MWRnet network (Cimini et al., 2012). Results showed neutral-to-positive impact. However, these experiments used retrieved variables (temperature and humidity profiles) whereas the assimilation of raw measurement (TBs) is found more optimal in the case of satellite data (Geer et al., 2008).

Accordingly, a potential way to increase the impact of MWR DA is to assimilate measured radiances (or brightness temperatures, TB) directly instead of retrieved profiles. With this type of assimilation, all the degrees of freedom for signal of MWR (Löhnert et al., 2009) can be used to improve the NWP model forecast in the PBL. In order to assimilate TB, a radiative transfer (RT) forward model is needed. The RT model allows to compute the TB for selected radiometer channels based on the NWP model state vector. TB differences between the modeled and measured observations can be used within a variational scheme (Courtier et al., 1998) that takes the corresponding uncertainties into account to retrieve temperature and humidity profiles in the first few kilometers from the ground, where the MWR provides the maximum information content. In addition, the Jacobians (i.e. partial derivatives with respect to the state vector) of the radiative transfer model are required to minimize the distances of the atmospheric state from both the first guess and the observations in a variational data assimilation process. These Jacobians represent the sensitivities of observations to the atmospheric thermodynamical state.

The fast RT model RTTOV (Radiative Transfer for the TIROS Operational Vertical Sounder (TOVS)) is widely used to simulate radiances from space-borne passive sensors. RTTOV has already been used for many years by many national meteorological services for assimilating down-looking observations from visible, infrared, and microwave radiometers, spectrometers and interferometers (Hocking et al., 2015 and references therein) aboard satellite platforms. The FORTRAN-90 code originally developed at ECMWF in the early 90's (Eyre, 1991) was intended for TOVS direct radiance assimilation within 3- and 4-dimentional variational analysis schemes (3DVAR, 4DVAR). Subsequently the original code has gone through several developments (e.g. Saunders et al.,





1999; Matricardi et al., 2001), more recently within the EUMETSAT NWPSAF, of which RTTOV v11.3 is the
latest version available. Since its first implementation and throughout its current version, RTTOV has been
developed and exploited for satellite observation perspective only. The model allows rapid simulations of
radiances for a suite of passive sensors given the atmospheric state vector, i.e. profiles of temperature, gas
concentration, cloud liquid water content and surface properties. The only one variable gas needed for RTTOV
v11 in the microwave band is water vapor. An important feature of RTTOV is that, in addition to the forward (or
direct) radiative transfer, it also computes the Jacobians, i.e. the gradient of the radiances with respect to the state
vector at the location in state space specified by the input state vector values. The Jacobians are calculated in
Tangent Linear (TL), Adjoint (AD) and K-modules of RTTOV.
There are other fast RT models used by NWP community for satellite data assimilation, like the Community
Radiative Transfer Model (CRTM - Ding et al., 2011). However, to our knowledge no fast RT model is currently
available to simulate ground-based radiometric observations. In this work, the version 11.2 of RTTOV has been
modified to handle ground-based microwave radiometer observations. The efforts for adapting RTTOV to
ground-based observations started within the COST action ES1202 (EG-CLIMET) and have been continued
within COST action ES1303 (TOPROF). The ground-based version of RTTOV developed here, called RTTOV-
gb, is able to simulate brightness temperatures from ground-based upward-looking microwave radiometers. In
addition, the TL, AD, and K-modules of RTTOV have been adapted to provide Jacobians for ground-based
geometry. We believe that the availability of RTTOV-gb with its K-module will enable more widespread and
better use of MWR observations in NWP models.
This paper introduces RTTOV-gb, the ground-based version of the fast radiative transfer model RTTOV. In
section 2 we describe the modifications made to the original RTTOV code for the ground-based radiative
transfer calculation. Section 3 discusses the performance of RTTOV-gb by evaluating its simulations against
those from accurate line-by-line RT models (3.1), against ground-based real MWR observations (3.2), against
analytic Jacobian calculations (3.3), and finally within an 1-dimentional variational (1D-Var) assimilation
scheme (3.4). Section 4 summarizes the findings and draws the final conclusions.
## 2    The formulation of the radiative transfer model
### 2.1    Radiative transfer model
Given a state vector $\mathbf{x}$ (the atmospheric thermodynamical state profile in radiative transfer problem), the radiance
vector (or brightness temperature) $\mathbf{y}$ is computed as:
$$\mathbf{y} = H(\mathbf{x}) \tag{1}$$
where H is the radiative transfer model (also referred to as the observation operator).
The core of RTTOV-gb simulates ground-based radiometer radiances using an approximated form of the
radiative transfer equation (RTE) for ground-based (upward-looking) observation geometry:
$$L_{ATM,i} = \tau_{i,toa} * B_i(T_{BKG}) + \int_{\tau_{i,toa}}^{1} B_i(T) d\tau \tag{2}$$
where $L_{ATM,i}$ is the radiance at the ground for channel i, neglecting scattering effects, $B_i$ is the Planck radiance at
channel i for a scene temperature T, $\tau_{i,toa}$ is the top-of-the-atmosphere transmittance and $T_{BKG}$ is the microwave



cosmic background temperature (2.728 K). Note that in the spectral range under consideration (20-60 GHz),
scattering is negligible for particles of the size of atmospheric molecules and cloud droplets, and even for larger
ice and snow particles (Kneifel et al., 2010). From a ground-based perspective, the transmittances and optical
depths are accumulated from the surface to the space instead of from the space to the surface as in the original
RTTOV satellite perspective. Consequently, several subroutines have been modified to reverse the accumulation
of transmittances and optical depths through the atmospheric path (see Section 6).
The RTE (2) is valid for both clear- and cloudy-sky conditions because in the microwave band RTTOV takes
into account the liquid water as an absorbing species and its effects are included through a contribution to the
transmittance profile. The first term of the right-hand side of the RTE (2) is the cosmic background radiation, the
second term is the atmospheric contribution.
The RTE (2) has been numerically solved over N atmospheric levels which are numbered from the top of the
atmosphere as follow:
- level $j = 1$, pressure $P_j = 0.005$hPa, temperature $T_j = T_1$ , transmittance $\tau_{ij} = \tau_{i,toa}$ for channel i;
- levels from $j = 2$ to $j = N-1$, $P_j$ are pressures for fixed levels, $\tau_{ij}$ is the surface-to-level transmittance for channel
i;
- level $j = M$, the first level which lies strictly above the input 2 m pressure (i.e. $M <= N$ and $P_M < P_{2m}$), $\tau_{ij} = \tau_{i,M}$
for channel i;
- level $j = N$, $P_N = 1050$hPa, surface air temperature $T_N = T_S$, $\tau_{iN} = 1$ for all channels;
For the ground-based perspective and each channel (omitting the i index for convenience), we define:

$$\begin{cases} \Delta\tau_j = \tau_{j+1} - \tau_j \\ \Delta B_j = B_j - B_{j+1} \\ \Delta d_j = d_j - d_{j+1} \end{cases} \tag{3}$$

where $\Delta d_j$ is the optical depth of the single layer j, $d_j$ is the level-to-surface optical depth.
The cosmic background radiation is:

$$L_{COSMIC} = \tau_1 * B(T_{BKG}) \qquad \text{with } \tau_1 = \tau_{toa} \tag{4}$$

The atmospheric contribution is:

$$L_A = \int_{\tau_{LEV=1}}^{\tau_M} B(T)d\tau + ST = \sum_{j=M}^{1}\left(\int_{\tau_j}^{\tau_{j+1}} B d\tau\right) + ST \tag{5}$$

where:

$$\int_{\tau_j}^{\tau_{j+1}} B d\tau = \tau_{j+1}B_{j+1} - \tau_j B_j + \frac{1}{\Delta d_j}\Delta B_j \Delta\tau_j = \Delta\tau_j * \left[B_{j+1} + \Delta B_j \frac{1}{\Delta d_j}\right] - \tau_j \Delta B_j \tag{6}$$

and ST is the contribution of the first layer above the surface:

$$ST = B_S(1 - \tau_M) - (B_M - B_S) + (B_M - B_S) * (1 - \tau_M) * \frac{1}{d_M} \tag{7}$$

with $B_S$ the Planck function evaluated at the input 2 m temperature.
In equation (6) we have used a parameterization of the Planck function (i.e. the so-called linear-in-tau
assumption, where tau means the optical depth of the single layer, corresponding to $\Delta d$ in the notation used in
this study). In the linear-in-tau assumption the source function throughout the layer is linear with the optical
depth of the layer (Saunders et al., 2010):
$$B[T(\Delta d)] = B_{j+1} + (B_j - B_{j+1})\frac{\Delta d}{\Delta d_j} \tag{8}$$





where $B_j$ is the Planck function for the top of the layer, $B_{j+1}$ is the Planck function at the bottom of the layer and
$\Delta d_j$ is the optical depth of the layer. In ground-based perspective $\Delta d$ goes from 0 to $\Delta d_j$ from the bottom to the
top of the layer.
The radiance for each channel i is then converted to an equivalent black-body temperature which is usually
called Brightness Temperature (TB) using the inverse Planck function.
**2.2    The input atmospheric profiles and near-surface variables**
The input profile data may be supplied on an arbitrary set of pressure levels. These consist of vertical profiles of
temperature (K) and humidity (ppmv) for clear-sky, and additional cloud liquid water content (CLW) profiles for
simulating cloudy conditions. In addition, pressure, temperature and humidity values at 2 m altitude are required.
The transmittance calculations described below are performed using atmospheric layers bounded by a number of
fixed pressure levels. RTTOV-gb interpolates the input profiles to the fixed pressure levels for the transmittance
calculation, but note that the RTE is integrated on the pressure levels supplied by the user (Hocking, 2014).
Currently RTTOV-gb uses fixed 101 pressure levels from 0.005hPa to 1050hPa for the transmittance calculation.
These levels have been specifically selected for ground-based perspective to be denser close to ground (34 levels
below 2km) than those usually used for the satellite perspective. Moreover they were chosen to improve the
accuracy of the optical depth prediction scheme used by RTTOV-gb compared to that obtained with the levels
used for satellite simulations. The vertical levels spacing is shown in Figure 1 in terms of level altitude
differences.
**2.3    Transmittance model**
The main variable computed in the radiative transfer model is the atmospheric optical depth for each channel i
and for each atmospheric layer j. The optical depths depend on the viewing angle of the instrument, pressure,
temperature, and concentrations of the absorbing species. The optical depths are computed by linear regression
from a set of predictors derived from the input state vector profile. Thus, the optical depth differences between
adjacent pressure levels are obtained through a polynomial in $X_{kj}$, the predictors, which are functions of
temperature T and specific humidity q at and above these levels (j being the level and k the number of predictors,
from 1 to P). The contribution of the water vapor on the optical depth is treated separately from that of
uniformly-mixed gases although they are calculated with two algorithms of the same form. The optical depth
from the surface to the level j in channel i along a path at an angle $\theta$ from the vertical, $d_{ij}$, is obtained as follow:
$$d_{ij} = d_{i,j+1} + \sum_{k=1}^{P} a_{ijk} X_{kj} \tag{9}$$
with $a_{ijk}$ the regression coefficients between optical depths and predictors.
There are three types of predictors for satellite perspective, predictors 7 (Matricardi et al., 2001), 8 (Matricardi,
2005) and 9 (Saunders, 2010), each of which is better suited for a specific application. The functional
dependence of the predictors used in RTTOV to parameterize the optical depths depends on factors such as the
absorbing gas, the angle $\theta$, the reference temperature and specific humidity profiles (the average of the training
profile set, respectively $T_j^{ref}$ and $q_j^{ref}$). Also the number of predictors depends on the selected gas.



We found the predictors 7 to give the best results for the ground-based geometry and thus they are used herewith
to train RTTOV-gb. The predictors 7 and the profile variables involved in the predictors calculation are listed in
Appendix A. Note that predictors 7 were originally developed for satellite simulations up to 60° zenith angles
and as such, the errors in the optical depth prediction increase for zenith angles above ~75º (i.e. for elevation
angles below ~15º). For MWR observations of the PBL thermodynamics, these scanning angles turn out to be
crucial, because of the information carried by opaque channels on the PBL temperature profile. Thus, it is
foreseen that an alternative set of predictors, specific for low elevation angles in the ground-based geometry,
may be worth investigating and developing in the future, though it is beyond the scope of this study.
The coefficients $a_{ikj}$ are calculated by linear regression of $\{d_{i,j} - d_{i,j+1}\}$ against $X_{kj}$. For the regression, $d_{ik}$ are
calculated using a line-by-line (LBL) model for a set of atmospheric profiles. Here we use 83 profiles from a
NWPSAF profile dataset interpolated on 101 pressure levels, already used for training RTTOV. It was
assembled to contain profiles representing the wide range of variation in humidity and temperature observed in
the Earth's atmosphere. Transmittances are computed for 6 optimally selected scanning angles which are
discussed in Section 3.1. LBL RT models provide accurate calculation of the atmospheric transmittances and
radiances, given the atmospheric profile of gas concentrations, and predefined spectral frequency grid. The LBL
optical depths must cover the full spectral range of all the radiometer channels of interest and provide a sufficient
resolution to represent accurately the transmittances in the channel spectral bands. The LBL model described by
Rosenkranz (1998; R98 herewith) has been used for gas absorption to calculate the clear-sky transmittances
needed in the RTTOV-gb regression coefficients computation. We limit the lowest elevation angle used in the
training phase to 10° because of the already mentioned limitation of the predictors 7.
If the optical depth for uniformly-mixed gases and water vapor are $d_{ij}^M$ and $d_{ij}^W$ respectively, the total optical
depth is:
$$d_{ij} = d_{ij}^M + d_{ij}^W \tag{10}$$
Then, optical depths are converted to transmittances:
$$\tau_{ij} = \exp(-d_{ij}) \tag{11}$$
Finally, RTTOV-gb computes the output radiances and TB from the derived transmittances and the input vertical
temperature profile using the radiative transfer equation (2).
**2.4    Jacobians: Tangent Linear, Adjoint and gradient matrix models**
The Jacobian matrix K gives the change in radiance $\boldsymbol{\delta y}$ for a change in any element of the state vector $\boldsymbol{\delta x}$
assuming a linear relationship about a given atmospheric state $\mathbf{x_0}$:
$$\boldsymbol{\delta y} = K(\mathbf{x_0})\,\boldsymbol{\delta x} \tag{12}$$
The elements of K contain the partial derivatives $\delta y_i/\delta x_j$, where the subscript i refers to channel and j to layer
number. The Jacobian provides the radiance sensitivity for each channel given unit perturbations at each level of
the profile vectors and in each of the surface parameters. It shows clearly, for a given profile, which layers in the
atmosphere are most sensitive to changes in temperature and variable gas concentrations for each channel. The
K-module of RTTOV computes the $K(\mathbf{x_0})$ matrix for each input profile. Alternatively, the Jacobians can be
computed with the so-called brute force (BF) method where K is estimated by perturbing each element of the



atmospheric state vector, repeating the RTTOV direct module iteratively. However, the calculations of the
Jacobians with the BF method are slower and less rigorous than with the K-module of RTTOV.
It is not always necessary to store and access the full matrix K; thus, the RTTOV package has routines to
compute the tangent linear only, i.e. the change in radiances $y_i$ for a given change in atmospheric profile $\boldsymbol{\delta x}$
around an initial atmospheric state $\mathbf{x_0}$.
$$\boldsymbol{\delta y}(\mathbf{x_0}) = \left[\boldsymbol{\delta x}\frac{\partial y_1}{\partial \mathbf{x}}, \boldsymbol{\delta x}\frac{\partial y_2}{\partial \mathbf{x}}, \boldsymbol{\delta x}\frac{\partial y_3}{\partial \mathbf{x}} \dots \dots \boldsymbol{\delta x}\frac{\partial y_{nchan}}{\partial \mathbf{x}}\right] \qquad \text{with } \frac{\partial}{\partial \mathbf{x}} = \boldsymbol{\nabla_x} = \left[\frac{\partial}{\partial x_1}, \frac{\partial}{\partial x_2}, \dots, \frac{\partial}{\partial x_N}\right] \qquad (13)$$
Similarly, the adjoint routines compute the change in any quantity of the state vector (e.g. T, q, surface variables
etc) $\boldsymbol{\delta x}$ around an assumed atmospheric state $\mathbf{x_0}$, given a change in the radiances $\boldsymbol{\delta y}$.
$$\boldsymbol{\nabla_x} = \boldsymbol{\nabla_x}\mathbf{y} \cdot \boldsymbol{\nabla_y} = K(\mathbf{x_0})^{\text{ T}} \cdot \boldsymbol{\nabla_y} \qquad (14)$$
$$\boldsymbol{\delta x}(\mathbf{x_0}) = \left[\boldsymbol{\delta y}\frac{\partial x_1}{\partial \mathbf{y}}, \boldsymbol{\delta y}\frac{\partial x_2}{\partial \mathbf{y}}, \boldsymbol{\delta y}\frac{\partial x_3}{\partial \mathbf{y}} \dots \dots \boldsymbol{\delta y}\frac{\partial x_N}{\partial \mathbf{y}}\right] \qquad (15)$$
For very large systems, it may be not feasible to calculate the full Jacobian matrix K and the tangent linear and
adjoint operations are computed instead.
The TL code is derived directly from the forward model because it represents the analytic derivative of the
radiance (forward model outputs) with respect to the atmospheric state vector **x**. The AD code is derived from
the TL by taking the transpose of the TL code. Finally, the K code is obtained from the AD code distributing the
AD level derivatives through the number of channels. Before running TL, AD, and K models, the direct model
needs to be run, because many of the intermediate variables calculated by the direct model are needed by the TL,
AD and K-modules.
**3    Performance of RTTOV-gb**
The performance of RTTOV-gb has been tested in four different ways, reported in the following sub-sections:
validation against the LBL RT model used as reference for the training and against another independent
reference LBL RT model (3.1); a comparison of TB simulated with RTTOV-gb from a radiosonde profile dataset
with nearly-colocated MWR measurements (3.2); a comparison of Jacobians calculated with the RTTOV-gb K-
module and the brute force method, and also with Jacobians computed with an analytical model (3.3);
exployment of RTTOV-gb as forward model operator within a 1-dimensional variational scheme (3.4).
**3.1    Comparison with line-by-line model computed radiances**
Brightness temperatures computed with RTTOV-gb have been compared against the corresponding TB
computed with the LBL model adopted for the regression training, i.e. the
R98 absorption model (Rosenkranz, 1998). For this purpose, we computed clear-sky TB with both, RTTOV-gb
and R98 at selected frequency channels from the set of 83 atmospheric profiles used in the training phase. Any
TB difference between the two models is assumed to be due to the regression error of RTTOV-gb with respect to
the reference LBL model. The analysis of the results focusses on the regression error in terms of the bias and
root-mean-square error (RMS) of the TB differences between RTTOV-gb and R98 simulations for each
considered channel. Here we consider 14 channels commonly used by commercial MWR, in particular the



Humidity And Temperature PROfiler (HATPRO, Rose et al., 2005): 22.24, 23.04, 23.84, 25.44, 26.24, 27.84,
31.40, 51.26, 52.28, 53.86, 54.94, 56.66, 57.30, and 58.00 GHz.
Using predictors 7, a sensitivity test to determine the optimal set of 6 elevation angles used for the computation
of regression coefficients for RTTOV-gb is shown in Table 1. The test has been made using 4 different sets of
elevation angles (i.e. 90-53-42-35-30-26°; 90-42-30-24-19-16°; 90-42-30-24-19-10°; and 90-42-30-19-10-5°).
The performance with respect to the R98 reference TB is presented for 4 elevation angles: 90, 30, 19 and 10°.
TB differences for all the training sets show bias and RMS lower than the manufacturer error specification for
HATPRO channels (~0.5 K - Rose et al., 2005) for all the tested elevation angles. The only exception are the
water vapor channels (22-31 GHz) at 10° elevation angle with the training sets 90-26°, 90-10° and 90-5°. This
result seems to confirm that predictors 7 are not ideal for elevation angles lower than 15°. However, it is
encouraging to note that even at 10°, bias and RMS are within the instrumental error for all the channels when
the training set 90-16° is adopted. Moreover, the agreement with the LBL model at low elevation angles is better
for the V-band opaque channels, which are most important for PBL temperature retrieval. Table 1 shows that the
training configuration optimizing the performance of RTTOV-gb is the set of elevation angles from 90° to 16°.
Somewhat surprising, this training configuration gives acceptable results even at 10° despite this elevation angle
being outside the training angle range.
Figure 2 shows an example of two spectra computed at HATPRO channels by RTTOV-gb and line-by-line
model R98 for the same atmospheric profile belonging to the dependent set. For this particular case, TB
differences between the two models are within 0.1K for all channels.
In Figure 3, statistics (bias, RMS and maximum value) of the TB difference between RTTOV-gb and LBL R98
for the dependent 83-profile set are shown. The optimal training configuration determined above (elevation
angles from 90° to 16°) is used. Results are shown for tests at four elevation angles (90, 30, 19 and 10°). At 90°
elevation, bias and RMS are respectively less than 0.030 K and 0.060 K for K-band (22-31GHz) and 0.003 K
and 0.025 K for the V-band opaque channels (54-58GHz). For these channels the maximum difference does not
exceed 0.15 K. The agreement is slightly worse at transparent V-band channels (51-54 GHz), with bias, RMS,
and maximum difference respectively within 0.03 K, 0.2 K, and 0.6 K. The larger discrepancies at transparent V-
band channels are probably due to the combined influence of temperature and water vapor, which likely
decreases the correlation of layer opacity with the two thermodynamical variables. Figure 3 shows that similar
results are found for other elevation angles, such as 30 and 19°. Note that these error statistics at 90° elevation
are about one order of magnitude larger than the analogous statistics of the original nadir-looking RTTOV
version (Saunders, 2002; Saunders, 2010) with a dependent profile set. We believe the reason is related to the
following. For the satellite (down-looking) case, if RTTOV underestimates the atmospheric optical depths (with
respect to reference LBL calculations), then too small atmospheric emission is computed and the atmospheric
contribution to the top-of-atmosphere radiance is too small. However, if the optical depths are underestimated
then the atmosphere as a whole is too transparent, so there will also be a larger contribution from the relatively
warmer lower atmosphere and/or surface. These two errors (too little contribution from the layers with
underestimated optical depth and too much contribution from the surface/lower atmosphere) have opposite sign
and tend to compensate for one another to some extent. Similarly, if RTTOV overestimates the optical depths,
there is more atmospheric emission but less contribution from the surface/lower atmosphere. Conversely, from
ground-based perspective the cosmic background (and the upper atmosphere) is relatively colder so there is no





compensation when the optical depths are either over- or underestimated. If optical depths are underestimated,
then generally smaller simulated radiances are computed (less atmospheric emission, more contribution from the
colder upper atmosphere). Vice versa, if optical depths are overestimated, higher simulated radiances are
computed (more atmospheric emission, less contribution from the colder upper atmosphere).
Note that Figure 3 shows bias, RMS, and maximum difference respectively up to -0.3, 0.4, 1.5 K for K-band
channels at 10° elevation. These are significantly larger compared to higher elevation angles. This is attributed to
the use of predictors 7, which are not optimal for elevation angles lower than 15°, and to the fact that 10° is
outside the elevation angle range used in the training configuration (90°-16°). However, extending the range of
training elevation angles to 10° or less generally degrades statistics as shown in Table 1. Nonetheless, we
highlight that the RMSs in Figure 3 are smaller than the uncertainty associated with TB observations (~0.5 K)
for all channels and all elevation angles.
Similarly, RTTOV-gb has been compared with the reference LBL model using an independent set of 52 profiles
(i.e. not used for training) on the same 101 pressure levels described earlier. These profiles are obtained by
interpolation from an NWPSAF profile set different from the one used to generate the dependent profile set. Test
results are presented in Figure 4 in terms of bias, RMS and maximum of the TB differences between RTTOV-gb
and the reference LBL R98. Results for the optimal training configuration and for elevation angles 90, 30, 19 and
10° are shown. Statistics are similar to those obtained with the dependent profile set. In this case however, the
error statistics are of the same order of magnitude than the analogous performance of the original nadir-looking
RTTOV with an independent profile set (Saunders, 2002; Saunders, 2010). For elevation angles down to 19°,
biases range from less than 0.002 K for the opaque channels to 0.020 K for K-band, while RMS is less than
0.060 K for K-band and 0.025 K for the opaque channels. The maximum TB differences do not exceed 0.5 K.
Similarly to the test with the dependent profile set, larger discrepancies are found in the transparent V-band
channels (51-54 GHz) and for K-band channels at 10° elevation. All the statistics obtained with the independent
profile set and the optimal training configuration are summarized in Table 2. Consistently with the dependent
test, the independent test in Figure 4 and Table 2 confirms that the RMSs are smaller than the uncertainty
associated with TB observations for all channels and all elevation angles.
The previous tests against the reference LBL R98 model have been performed also at the 22 frequency channels
(22-60 GHz) used by another commercial MWR, the MP-3000A (Cimini et al., 2011). Statistics, reported in
Table 3 in terms of bias and RMS, are similar to those obtained for HATPRO channels, at both K- and V-band.
It is worth remembering that R98 is the model used to train the regression scheme. In order to perform a
completely independent test, we compare RTTOV-gb with an independent reference radiative transfer model, the
Atmospheric Radiative Transfer Simulator (ARTS, Buehler et al. 2005; Eriksson et al., 2011; Eriksson et al.,
2015) and a completely different profile dataset. In this test, HATPRO observations are simulated using
RTTOV-gb and ARTS from a set of 1327 profiles from the AROME analysis. AROME is the French convective
scale NWP model with a 2.5 km horizontal grid mesh developed by Météo-France (Seity et al., 2010). The
analysis dataset consists of 60-level pressure, temperature, specific humidity and liquid water content profiles
over Bordeaux (SouthWest of France, latitude 44.498°N, longitude 0.418°E, elevation 49 m) from April to
October 2014. Both clear- and cloudy-sky conditions are considered. This dataset, which is limited in space,
time, and thus in atmospheric conditions, was chosen to demonstrate the performance of RTTOV-gb in typical
deployment environment. In this analysis, ARTS settings for absorption model have been selected to adopt as





much as possible the same absorption model as RTTOV-gb: R98 for oxygen and water vapor absorption, and the
model described in Liebe et al. (1993) for cloud liquid water (referred as MPM93 within ARTS). Note that
MPM93 is the only option for liquid water absorption available in ARTS. Conversely RTTOV-gb is consistent
with the original RTTOV, which adopts a combination of Liebe et al. (1991) and Lamkaouchy et al. (1997)
models (English et al., 1999).
This comparison is presented in Figure 5 in terms of bias, standard deviation (std), and RMS of TB differences,
at 90° elevation angle. Here we have discarded TB differences that are larger than 3 std from the mean (21
profiles out of 1327). Bias less than 0.18 K for K-band and less than 0.08 K for opaque V-band channels are
shown. RMS and standard deviation are close, ranging from 0.1-0.25 K for K-band channels, and within 0.1 K
for V-band opaque channels (55-58 GHz). Similar to previous tests, larger discrepancies are found in the more
transparent V-band channels (51-54 GHz) with a RMS error up to 0.5 K at 51 GHz in cloudy-sky. Comparing
Figures 4 and 5, we notice slightly larger statistics (by 0.1-0.2 K) in the RTTOV-gb vs. ARTS than in the
RTTOV-gb vs. R98 tests. We attribute this to the fact that RTTOV-gb is totally independent of ARTS and
moreover to the specific profile dataset, which likely introduces biases with respect to the RTTOV-gb training
climatology. Note that TB differences for all the channels are of the same order of magnitude of those found
between ARTS and the original nadir-looking RTTOV (Buehler et al., 2006). This demonstrates comparable
capabilities between RTTOV-gb and the original version of RTTOV. RMS TB differences between RTTOV-gb
and ARTS at 90° elevation are within 0.5 K, thus below the uncertainty associated with TB observations. From
the three tests above, we can conclude that in the elevation angle range from 90 to 10° the forward model error
due to the use of the fast RT with respect to the reference LBL model is within the instrument uncertainty. This
confirms that RTTOV-gb can be safely deployed in place of a LBL model into variational assimilation schemes.
**3.2    Comparison with real observations**
Another way to evaluate RTTOV-gb is to compare TB simulated from radiosonde profiles with TB measured by
a nearly colocated MWR. This comparison provides an end-to-end evaluation of the model, though radiosonde
drift, MWR calibration, finite beamwidth, discretization, and instrumental noise all contribute to the total
uncertainty. Nevertheless, observations minus background model (O-B) differences are the primary input for
direct radiance assimilation into a NWP model and thus need to be investigated and understood. For this
analysis, we exploit a dataset of 365 radiosonde profiles collected over Bordeaux from April to October 2014,
together with the nearly simultaneous TB observed by a ground-based MWR (HATPRO) operated at the
radiosonde launching site. The dataset was first reduced to clear-sky conditions. To be conservative, clear-sky
conditions have been selected using a three-fold screening, based on (i) ceilometer Cloud Base Height (CBH),
(ii) sky infrared temperature ($T_{IR}$), and (iii) 20-minute standard deviation of liquid water path ($\sigma_{LWP}$) from
HATPRO. Thus, periods with CBH below maximum range (8000m), $T_{IR}$>-30°C, or $\sigma_{LWP}$ >$10^{-2}$ kg/m$^3$ were
rejected. Moreover, cases with integrated water vapor differences between MWR and radiosonde profiles larger
than 1 mm have been discarded in order to reduce instrumental uncertainties involved in the comparison. After
this screening, only 23 profiles remained for the analysis. Bias, standard deviation, and RMS differences
between TB observed by the MWR and simulated with both RTTOV-gb and ARTS are shown in Figure 6. With
respect to the MWR observations, RTTOV-gb shows bias from 0.02 K at 22.24 GHz to 0.5 K at 23.84 GHz in



the K-band and from 0.16 K to 0.31 K in the V-band opaque channels. RMS range from 0.90 K to 0.47 K in the
K-band and from 0.41K to 0.64K in the V-band opaque channels. Larger bias is found at V-band transparent
channels: 1-2 K at 51.26 GHz and 4-5 K at at 52.28GHz with either RTTOV-gb or ARTS simulations. Note that
RTTOV-gb and ARTS show similar statistics with respect to MWR observations. This result is very important
as it suggests that forward model errors due to the fast model approximation are not dominant. Note that bias
values of the same order of magnitude for the 51-54 GHz range were previously reported (Hewison et al. 2006;
Löhnert and Maier, 2012, Martinet et al 2015, Blumberg et al., 2015), employing MWR of different types and
manufacturers. This may be attributed to a combination of uncertainties from instrument calibration and gas
absorption models. In fact, semi-transparent channels (as in the 51-54 GHz range) suffer from larger calibration
uncertainties due to the lack of a close reference-temperature calibration point. In addition, their response is
influenced by the water vapour continuum and oxygen line coupling, which contribute significantly to the
uncertainties because their parametrization is extrapolated from laboratory measurements to typical atmospheric
conditions. It is beyond the scope of this paper to investigate spectroscopy issues, but our results support
previous evidence and point to the need for further lab measurements, e.g. Boukabara et al (2005). Considering
that O-B systematic differences are usually evaluated and removed before assimilating data into NWP, we
believe that statistics in Figure 6 support the safe use of RTTOV-gb for direct radiance assimilation of MWR TB
into NWP models.
### 3.3    Comparison of Jacobians
After testing the RTTOV-gb direct module, the RTTOV-gb Jacobians calculation needs to be tested in order to
provide a complete tool for a fast and safe MWR data assimilation. First, a consistency test of the Jacobians
calculated with TL-, AD- and K-modules of RTTOV-gb has been performed to ensure the correctness of the
TL/AD/K coding modified for ground-based perspective. The test resulted in nearly the same Jacobians for TL,
AD- and K-modules. Subsequently, the temperature and humidity Jacobians calculated with RTTOV-gb K-
module have been compared with those computed with the brute force (BF) method for a specific cloudy sky
profile. The BF method calculates the Jacobian by finite differences by calling the direct module multiple times
after perturbing each individual input profile variable. The consistency of K-module with BF was confirmed
using the RTTOV test suite (Brunel et al., 2014), bearing in mind that some small differences between the
Jacobians are expected. Figure 7 shows the temperature and absolute humidity Jacobians for the V- and K-bands
channels. The Jacobians computed with RTTOV-gb BF and K-module are almost identical with differences
smaller than 1%. As expected, the TB sensitivity to atmospheric temperature is higher in the low troposphere,
especially in the PBL, and it increases with frequency in the spectral range between 51 and 58 GHz. Between 22
and 31 GHz, the sensitivity of the TBs to water vapor is almost independent of altitude and decreases with
increasing frequency.
The Jacobians for cloud liquid water (CLW) are needed when cloudy-sky conditions are considered. Figure 8
shows a comparison of CLW Jacobians calculated with RTTOV-gb K-module and BF method. Similar to
temperature and humidity, they are found to be almost identical (differences smaller than 0.1%, likely due to
truncation errors). As expected, the TB sensitivity to CLW increases with frequency in the K-band, while it
decreases with frequency in the V-band due to the increasingly dominant oxygen absorption. TB are sensitive to



CLW at all levels up to 322hPa (about 10km), where RTTOV, and thus also RTTOV-gb, have set their upper
limit for non-zero CLW.
For a completely independent test, Jacobians calculated with RTTOV K-module have been compared with those
computed with the reference radiative transfer model ARTS. To that end the Qpack2 package (Eriksson et al.,
2005) provided with the ARTS software was used. ARTS Jacobians are derived from a semi-analytical
expression described in Eriksson and Buehler, 2015. As shown in Figure 9, temperature and humidity Jacobians
from ARTS and RTTOV-gb are found to be almost identical, either for K-band and V-band channels, with
differences smaller than 3% for temperature and 5% for humidity. Figure 10 shows the comparison of CLW
Jacobians from ARTS and RTTOV-gb. These are similar to each other, both in shape and order of magnitude,
from surface up to 322 hPa (RTTOV cloud limit). However, differences of about 10% occur around 450 hPa,
particularly at transparent channels (31, 51 and 52 GHz). These are likely due to small differences in the liquid
water absorption models in ARTS and in RTTOV-gb, as mentioned above in Section 3.1. However, for a typical
CLW profile, these model differences lead to small TB differences (order of 0.1 K) and are thus deemed as
negligible.
**3.4    1D-Var application**
Finally, RTTOV-gb has been tested as forward model within a 1-dimensional variational (1D-Var) scheme. For
this purpose, the 1D-Var software package provided by the NWPSAF (Weston, 2014) has been adapted in the
framework of the COST Action TOPROF to exploit RTTOV-gb. Among other modifications, the 1D-Var tool
has been modified to allow the assimilation of observations at different elevation angles for the same instrument.
The 1D-Var approach searches the atmospheric state $\mathbf{x}$ that minimizes both the distance to the background $\mathbf{x_b}$ and
the observation $\mathbf{y}$. The cost function J needs to be minimized modifying the different variables defined in the
control vector $\mathbf{x}$ (Cimini et al., 2010):
$$J = \frac{1}{2}[\mathbf{y} - H(\mathbf{x})]^T R^{-1}[\mathbf{y} - H(\mathbf{x})] + \frac{1}{2}[\mathbf{x} - \mathbf{x_b}]^T B^{-1}[\mathbf{x} - \mathbf{x_b}]$$    (16)
Here B represents the background-error covariance matrix and R the observation error covariance matrix. H
represents the observation operator, in our case RTTOV-gb. The background profile comes from a short-range
forecast of a NWP model or from a colocated radiosonde. Here, $\mathbf{x_b}$ is a 3-hour forecast from the French
convective scale model AROME. The Jacobians needed to minimize the cost function J are calculated with the
RTTOV-gb K-module.
The aim is to retrieve temperature and humidity profiles and column-integrated liquid water path from MWR
observations through a 1D-Var retrieval approach exploiting RTTOV-gb. To this aim, an Observing-System
Simulation Experiment (OSSE) was set up with 224 AROME analyses profiles in February 2015 over the Alps
with the new horizontal grid mesh of 1.3 km. These analyses are made of 90-level pressure, temperature, specific
humidity and liquid water content profiles, typical of an alpine valley and mountainous region in winter. Both
clear- and cloudy-sky conditions are considered. Starting from the AROME unperturbed profiles (the "truth"),
background profiles are created by perturbing the initial AROME profiles according to the background error
covariance matrix B. In this study, the B matrix was computed from an AROME ensemble assimilation system
following the approach used to derive this matrix operationally at Météo-France (Brousseau et al., 2011). By
applying RTTOV-gb to the unperturbed AROME, observations are created by adding synthetic observation





errors to the RTTOV-gb simulations. The synthetic random errors are assumed to follow a diagonal R matrix
with reasonable standard deviations, i.e. ~ 0.2-1.0 K depending on channels (Hewison, 2007).
In clear-sky conditions, temperature and specific humidity are used as control variables in the 1D-Var. A
comparison between temperature and humidity retrievals obtained with 1D-Var, the corresponding unperturbed
and background profiles for two retrieval examples are shown in Figure 11. As expected, the 1D-Var retrievals
are closer to the "truth" than the background profiles. In this case 1-D-Var provides an improvement with respect
to the background in the first 2 km for temperature and in the first 4 km for humidity, which is encouraging for
future data assimilation experiments. A comprehensive evaluation of RTTOV-gb plus 1D-Var for data
assimilation using real MWR observations will be subject of future work.
Here, we just underline that the main advantage of RTTOV-gb with respect to LBL models is the considerably
lower computation time. RTTOV-gb direct module is found to be 5 times faster than the line-by-line model
ARTS (Martinet et al. 2015). The RTTOV-gb K-module is found to be 10 times faster than ARTS Jacobians
calculation. Moreover, our tests demonstrate that the computation time for Jacobians is shorter by a factor of 8
for RTTOV-gb K-module than for direct module with brute force method.

## 4    Summary

The version 11.2 of the fast radiative transfer model RTTOV, developed for space-borne sensors, has been
successfully modified to simulate ground-based microwave radiometer observations. In addition to the direct
module, which allows to simulate ground-based MWR observations, the TL-, AD- and K-modules of RTTOV
have been modified in order to provide temperature, humidity and cloud liquid water Jacobians for the ground-
based perspective. We introduced the ground-based version of RTTOV, called RTTOV-gb, and demonstrated its
potential for fast MWR TB simulations from thermodynamic profiles. RTTOV-gb has been validated against
accurate, but less time-efficient, reference line-by-line models and real MWR observations. Results demonstrate
its applicability as a forward model within a variational scheme for fast and safe MWR data assimilation into
NWP models. It is believed that the direct assimilation of TB, instead of retrieved profiles, may improve the
impact of MWR observations for temperature and humidity profiles analysis in the first few kilometers from the
ground, where MWR provides the maximum information content.
The performance of RTTOV-gb has been validated by comparison with TB simulated with the line-by-line
model R98 (Rosenkranz, 1998), the same model as used for the RTTOV training phase. For both dependent and
independent profile sets, RMS are below the typical TB uncertainty of ground-based MWR (~0.5 K) ranging
from a maximum of 0.06 K for the water vapor band to 0.025 K for the V-band opaque channels. Larger
discrepancies are observed at the transparent V-band channels (51 and 52 GHz), with RMS within 0.20 K, and at
elevation angle 10°. TB simulated with RTTOV-gb from AROME analyses have also been compared with those
simulated with the reference line-by-line model ARTS. At 90° elevation, for both clear- and cloudy-sky
conditions TB differences do not exceed 0.25 K in terms of biases and RMS at all HATPRO channels except for
the transparent V-band channels 51-52 GHz (up to 0.5 K in cloudy-sky conditions). Finally, RTTOV-gb has
been validated by radiosonde-derived TB with real nearly collocated MWR observations. In this case RMS
increases with respect to the RTTOV-gb/LBL comparisons ranging from 0.90 K to 0.47 K in the K-band and
from 0.41 K to 0.64 K in the V-band opaque channels. Larger discrepancies were found at V-band transparent





channels, which may be explained by calibration and gas absorption uncertainties. However, the statistics of
RTTOV-gb and ARTS simulations with respect to MWR observations are similar for each channel, suggesting
that forward model errors due to the fast model approximation are not dominant. Temperature, humidity and
cloud liquid water Jacobians computed with RTTOV-gb K-modules were found to be similar in shape and
magnitude with those calculated with the brute force method or with the ARTS model.
Finally, RTTOV-gb has been tested as a forward model within a 1D-Var software package in an OSSE to
improve AROME thermodynamic profiles estimated by directly assimilating synthetic MWR TB. For both
temperature and humidity profiles the 1D-Var considerably improves the retrievals with respect to the
background, in the first few kilometers from the ground. Concerning the computation speed, RTTOV-gb with K-
module is found to be 8 times faster in computing Jacobians than the brute force method. RTTOV-gb direct
module is found to be 5 times faster than the radiative transfer model ARTS and RTTOV-gb K-module is found
to be 10 times faster than ARTS Jacobians calculation.
Ultimately, this analysis confirms that RTTOV-gb is able to correctly simulate ground-based MWR radiances
and to reproduce reasonable temperature, humidity and cloud liquid water Jacobians. In conclusion RTTOV-gb
is well suited for serving as forward model in a variational data assimilation scheme for a direct, safe, and fast
NWP data assimilation of real MWR radiance observations.

## 5    Acknowledgements

This work has been stimulated through the COST Action ES1303 (TOPROF), supported by COST (European
Cooperation in Science and Technology). The authors would like to acknowledge the NWPSAF and Met Office,
in particular Peter Rayer, for providing support on RTTOV coding and the Meteo-France for providing AROME
analyses and measurements performed in the Bordeaux campaigns.

## 6    Code and data availability

The original RTTOV v11.2 can be obtained via the request form in the NWPSAF web site
(http://nwpsaf.eu/site/software/rttov/rttov-v11/).
The efforts for adapting RTTOV to ground-based observations started within the COST (http://www.cost.eu/)
action ES1202 (EG-CLIMET) and have been continued within COST action ES1303 (TOPROF,
http://www.toprof.eu/). The modifications needed to adapt the radiative transfer equation from satellite to the
ground-based perspective have been made in the subroutine src/main/rttov_integrate.F90. The RTTOV
subroutines that have been modified in RTTOV-gb to reverse the way to initialize and accumulate transmittances
and optical depths are respectively src/main/rttov_transmit.F90 and src/main/rttov_opdep.F90. The calculation of
the predictors 7 for the ground-based perspective have been adapted in the subroutine
src/main/rttov_profaux.F90. Modifications made in the direct module of RTTOV v11.2 code have been imported
in the corresponding TL-, AD- and K-modules subroutines (i.e. rttov_integrate_tl.F90, rttov_integrate_ad.F90,
rttov_integrate_k.F90; rttov_transmit_tl.F90, rttov_transmit_ad.F90, rttov_transmit_k.F90; rttov_opdep_tl.F90,
rttov_opdep_ad.F90, rttov_opdep_k.F90). The conditions of release of RTTOV-gb are currently under
discussion among NWPSAF and COST action TOPROF.





All the informations needed to download the ARTS code can be found in the web site
http://www.radiativetransfer.org/.
The NWPSAF profiles, from which we interpolated the profile sets used for the RTTOV-gb training and
independent test, are available at https://nwpsaf.eu/deliverables/rtm/profile_datasets.html.
The AROME analyses used for ARTS/RTTOV-gb comparison and 1D-Var application, and the
MWR/radiosondes dataset used for the validation against real MWR measurement can be obtained by email to
pauline.martinet@meteo.fr.
**Appendix A**
The predictors $X_{kj}$ introduced in Section 2 are functions of the absorbing gas, the zenith angle $\theta$, the pressure,
temperature and water vapor mixing ratio profiles, and finally the reference temperature and water vapor mixing
ratio profiles (i.e. the average of the training profile set). These are defined in Matricardi et al. (2001) and briefly
summarized below. Introducing at each fixed level $j$ the pressure $P^{prof}(j)$, the temperature and the water vapor
mixing ratio $T^{prof}(j)$ and $W^{prof}(j)$, and the corresponding reference $T^{ref}(j)$ and $W^{ref}(j)$, the following variables
are defined:
$T(j) = \left[T^{prof}(j) + T^{prof}(j+1)\right]/2$
$T^*(j) = \left[T^{ref}(j) + T^{ref}(j+1)\right]/2$

$W(j) = \left[W^{prof}(j) + W^{prof}(j+1)\right]/2$

$W^*(j) = \left[W^{ref}(j) + W^{ref}(j+1)\right]/2$

$P(j) = \left[P^{prof}(j) + P^{prof}(j+1)\right]/2$

$T_r(j) = T(j)/T^*(j)$
$\delta T(j) = T(j) - T^*(j)$

$W_r(j) = W(j)/W^*(j)$

$T_w(j) = \sum_{l=N-1}^{j} P(l+1)[P(l+1) - P(l)]T_r(l+1)$ with $T_w(j=N) = 0$ at the surface.

$$W_w(j) = \left\{\sum_{l=N-1}^{j} P(l+1)[P(l+1) - P(l)]W(l)\right\} / \left\{\sum_{l=N-1}^{j} P(l+1)[P(l+1) - P(l)]W^*(l)\right\}$$

The RTTOV predictors 7 are derived from the variables above as listed in Table A1.

| Predictor 7 | Mixed Gases | Water Vapor |
|:---:|:---:|:---:|
| $X_{1,j}$ | $\sin(\theta)$ | $\sin^2(\theta)W_r^2(j)$ |
| $X_{2,j}$ | $\sin^2(\theta)$ | $(\sin(\theta)W_w(j))^2$ |
| $X_{3,j}$ | $\sin(\theta)T_r(j)$ | $(\sin(\theta)W_w(j))^4$ |
| $X_{4,j}$ | $\sin(\theta)T_r^2(j)$ | $\sin(\theta)W_r(j)\delta T(j)$ |
| $X_{5,j}$ | $T_r(j)$ | $\sqrt{\sin(\theta)W_r(j)}$ |
| $X_{6,j}$ | $T_r^2(j)$ | $\sqrt[4]{\sin(\theta)W_r(j)}$ |
| $X_{7,j}$ | $\sin(\theta)T_w(j)$ | $\sin(\theta)W_r(j)$ |
| $X_{8,j}$ | $\sin(\theta)\dfrac{T_w(j)}{T_r(j)}$ | $(\sin(\theta)W_r(j))^3$ |





| | | |
|---|---|---|
| $X_{9,j}$ | $\sqrt{\sin(\theta)}$ | $(\sin(\theta)W_r(j))^4$ |
| $X_{10,j}$ | $\sqrt{\sin(\theta)}\sqrt[4]{T_w(j)}$ | $\sin(\theta)\,W_r(j)\delta T(j)|\delta T(j)|$ |
| $X_{11,j}$ | $0$ | $\left(\sqrt{\sin(\theta)}W_r(j)\right)\delta T(j)$ |
| $X_{12,j}$ | $0$ | $\dfrac{(\sin(\theta)W_r(j))^2}{W_w}$ |
| $X_{13,j}$ | $0$ | $\dfrac{\sqrt{\sin(\theta)}\,W_r(j)W_r(j)}{W_w(j)}$ |
| $X_{14,j}$ | $0$ | $\sin(\theta)\dfrac{W_r^2(j)}{T_r(j)}$ |
| $X_{15,j}$ | $0$ | $\sin(\theta)\dfrac{W_r^2(j)}{T_r^4(j)}$ |

**Table A1: Predictors 7 used for mixed gases and water vapor (after Matricardi et al. 2001).**

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





|  |  | BIAS (K) | | | | RMS (K) | | | |
|---|---|---|---|---|---|---|---|---|---|
|  |  | Elevation angle 90° | | | | | | | |
| CHAN # | Frequency(GHz) | 90°-26° | 90°-16° | 90°-10° | 90°-5° | 90°-26° | 90°-16° | 90°-10° | 90°-5° |
| 1 | 22.24 | -0.007 | -0.016 | 0.004 | -0.170 | 0.033 | 0.059 | 0.047 | 0.373 |
| 2 | 23.04 | -0.002 | -0.009 | 0.011 | -0.159 | 0.029 | 0.052 | 0.050 | 0.352 |
| 3 | 23.84 | 0.005 | 0.002 | 0.023 | -0.132 | 0.028 | 0.043 | 0.053 | 0.308 |
| 4 | 25.44 | 0.009 | 0.011 | 0.029 | -0.087 | 0.029 | 0.036 | 0.056 | 0.224 |
| 5 | 26.24 | 0.009 | 0.011 | 0.028 | -0.074 | 0.029 | 0.035 | 0.054 | 0.195 |
| 6 | 27.84 | 0.008 | 0.012 | 0.025 | -0.059 | 0.029 | 0.034 | 0.050 | 0.158 |
| 7 | 31.40 | 0.009 | 0.011 | 0.023 | -0.049 | 0.033 | 0.038 | 0.049 | 0.128 |
| 8 | 51.26 | 0.017 | 0.024 | 0.043 | -0.101 | 0.175 | 0.176 | 0.159 | 0.244 |
| 9 | 52.28 | 0.021 | 0.025 | 0.039 | -0.070 | 0.202 | 0.201 | 0.186 | 0.246 |
| 10 | 53.86 | 0.010 | 0.012 | 0.015 | 0.001 | 0.116 | 0.118 | 0.115 | 0.122 |
| 11 | 54.94 | 0.002 | 0.003 | 0.004 | -0.008 | 0.023 | 0.023 | 0.023 | 0.023 |
| 12 | 56.66 | 0.001 | 0.001 | 0.001 | 0.001 | 0.007 | 0.007 | 0.007 | 0.007 |
| 13 | 57.30 | 0.001 | 0.001 | 0.001 | 0.001 | 0.005 | 0.005 | 0.005 | 0.005 |
| 14 | 58.00 | 0.000 | 0.001 | 0.001 | 0.001 | 0.004 | 0.004 | 0.004 | 0.004 |

|  |  | Elevation angle 30° | | | | | | | |
|---|---|---|---|---|---|---|---|---|---|
| CHAN # | Frequency(GHz) | 90°-26° | 90°-16° | 90°-10° | 90°-5° | 90°-26° | 90°-16° | 90°-10° | 90°-5° |
| 1 | 22.24 | 0.002 | 0.027 | 0.020 | 0.036 | 0.033 | 0.047 | 0.046 | 0.180 |
| 2 | 23.04 | 0.000 | 0.025 | 0.019 | 0.029 | 0.030 | 0.047 | 0.043 | 0.173 |
| 3 | 23.84 | -0.002 | 0.020 | 0.016 | 0.014 | 0.026 | 0.040 | 0.040 | 0.162 |
| 4 | 25.44 | -0.004 | 0.013 | 0.013 | -0.007 | 0.024 | 0.037 | 0.037 | 0.150 |
| 5 | 26.24 | -0.004 | 0.010 | 0.012 | -0.012 | 0.023 | 0.036 | 0.035 | 0.145 |
| 6 | 27.84 | -0.003 | 0.008 | 0.011 | -0.016 | 0.024 | 0.037 | 0.033 | 0.137 |
| 7 | 31.40 | -0.004 | 0.006 | 0.010 | -0.019 | 0.155 | 0.043 | 0.036 | 0.131 |
| 8 | 51.26 | 0.010 | 0.018 | 0.027 | -0.079 | 0.029 | 0.171 | 0.162 | 0.211 |
| 9 | 52.28 | 0.016 | 0.019 | 0.026 | -0.073 | 0.138 | 0.149 | 0.143 | 0.174 |
| 10 | 53.86 | 0.003 | 0.007 | 0.008 | -0.005 | 0.026 | 0.028 | 0.028 | 0.027 |
| 11 | 54.94 | 0.001 | 0.001 | 0.001 | -0.001 | 0.007 | 0.007 | 0.007 | 0.007 |
| 12 | 56.66 | 0.000 | 0.000 | 0.000 | -0.000 | 0.002 | 0.002 | 0.002 | 0.002 |
| 13 | 57.30 | 0.000 | 0.000 | 0.000 | -0.000 | 0.001 | 0.001 | 0.001 | 0.001 |
| 14 | 58.00 | 0.000 | 0.000 | 0.000 | -0.000 | 0.001 | 0.001 | 0.001 | 0.001 |

|  |  | Elevation angle 19° | | | | | | | |
|---|---|---|---|---|---|---|---|---|---|
| CHAN # | Frequency(GHz) | 90°-26° | 90°-16° | 90°-10° | 90°-5° | 90°-26° | 90°-16° | 90°-10° | 90°-5° |
| 1 | 22.24 | -0.050 | -0.004 | -0.065 | 0.203 | 0.078 | 0.044 | 0.086 | 0.317 |
| 2 | 23.04 | -0.053 | -0.005 | -0.070 | 0.189 | 0.079 | 0.042 | 0.089 | 0.298 |
| 3 | 23.84 | -0.056 | -0.007 | -0.074 | 0.158 | 0.083 | 0.038 | 0.090 | 0.259 |
| 4 | 25.44 | -0.046 | -0.007 | -0.070 | 0.099 | 0.089 | 0.036 | 0.087 | 0.192 |
| 5 | 26.24 | -0.039 | -0.006 | -0.066 | 0.080 | 0.089 | 0.036 | 0.084 | 0.171 |
| 6 | 27.84 | -0.028 | -0.005 | -0.059 | 0.055 | 0.091 | 0.036 | 0.078 | 0.149 |
| 7 | 31.40 | -0.018 | -0.004 | -0.052 | 0.035 | 0.103 | 0.043 | 0.077 | 0.139 |
| 8 | 51.26 | 0.020 | 0.013 | -0.018 | -0.003 | 0.139 | 0.128 | 0.132 | 0.152 |
| 9 | 52.28 | -0.031 | 0.012 | -0.004 | 0.021 | 0.085 | 0.085 | 0.085 | 0.097 |
| 10 | 53.86 | 0.004 | 0.001 | -0.000 | 0.003 | 0.013 | 0.010 | 0.010 | 0.011 |
| 11 | 54.94 | 0.002 | 0.000 | 0.000 | 0.001 | 0.005 | 0.003 | 0.003 | 0.004 |
| 12 | 56.66 | 0.000 | 0.000 | 0.000 | 0.000 | 0.001 | 0.001 | 0.001 | 0.001 |
| 13 | 57.30 | 0.000 | 0.000 | 0.000 | 0.000 | 0.000 | 0.000 | 0.000 | 0.000 |
| 14 | 58.00 | 0.000 | 0.000 | 0.000 | 0.000 | 0.000 | 0.000 | 0.000 | 0.000 |



| | | BIAS (K) | | | | RMS (K) | | | |
|---|---|---|---|---|---|---|---|---|---|
| | | | | Elevation angle 10° | | | | | |
| CHAN # | Frequency(GHz) | 90°-26° | 90°-16° | 90°-10° | 90°-5° | 90°-26° | 90°-16° | 90°-10° | 90°-5° |
| 1 | 22.24 | -0.299 | -0.324 | **-0.626** | **-0.930** | 0.428 | 0.381 | **0.681** | **1.035** |
| 2 | 23.04 | -0.297 | -0.317 | **-0.632** | **-0.955** | 0.461 | 0.369 | **0.685** | **1.027** |
| 3 | 23.84 | -0.391 | -0.312 | **-0.648** | **-0.998** | **0.662** | 0.356 | **0.698** | **1.067** |
| 4 | 25.44 | **-0.544** | -0.294 | **-0.664** | **-1.055** | **1.214** | 0.343 | **0.716** | **1.128** |
| 5 | 26.24 | **-0.573** | -0.284 | **-0.663** | **-1.065** | **1.414** | 0.342 | **0.718** | **1.143** |
| 6 | 27.84 | **-0.592** | -0.270 | **-0.659** | **-1.075** | **1.685** | 0.349 | **0.716** | **1.159** |
| 7 | 31.40 | **-0.594** | -0.260 | **-0.680** | **-1.129** | **2.023** | 0.377 | **0.731** | **1.205** |
| 8 | 51.26 | 0.000 | -0.088 | -0.337 | **-0.609** | 0.272 | 0.103 | 0.350 | **0.633** |
| 9 | 52.28 | -0.021 | -0.029 | -0.106 | -0.202 | 0.083 | 0.034 | 0.112 | 0.214 |
| 10 | 53.86 | 0.022 | 0.000 | -0.007 | -0.014 | 0.037 | 0.003 | 0.011 | 0.021 |
| 11 | 54.94 | 0.005 | 0.000 | -0.002 | -0.004 | 0.009 | 0.001 | 0.003 | 0.006 |
| 12 | 56.66 | 0.000 | 0.000 | -0.000 | -0.000 | 0.001 | 0.000 | 0.000 | 0.001 |
| 13 | 57.30 | 0.000 | 0.000 | -0.000 | -0.000 | 0.000 | 0.000 | 0.000 | 0.001 |
| 14 | 58.00 | 0.000 | 0.000 | -0.000 | -0.000 | 0.000 | 0.000 | 0.000 | 0.000 |

**Table 1: Statistics for the comparison between RTTOV-gb and the line-by-line model R98 (Rosenkranz, 1998) at**
**elevation angles 90, 30, 19 and 10°. The HATPRO channel number (CHAN #), the channel central frequency, bias and**
**RMS for each RTTOV training configuration are reported. The values which are larger than 0.5 K are highlighted in**
**bold.**

**TRAINING CONFIGURANTION: Elevation angles from 90° to 16°**

| | | BIAS (K) | | | | RMS (K) | | | |
|---|---|---|---|---|---|---|---|---|---|
| CHAN # | Frequency(GHz) | 90° | 30° | 19° | 10° | 90° | 30° | 19° | 10° |
| 1 | 22.24 | -0.008 | 0.021 | -0.004 | -0.282 | 0.049 | 0.045 | 0.042 | 0.326 |
| 2 | 23.04 | -0.002 | 0.020 | -0.006 | -0.276 | 0.042 | 0.045 | 0.042 | 0.319 |
| 3 | 23.84 | 0.007 | 0.017 | -0.008 | -0.273 | 0.035 | 0.044 | 0.045 | 0.320 |
| 4 | 25.44 | 0.018 | 0.001 | -0.009 | -0.257 | 0.032 | 0.042 | 0.051 | 0.339 |
| 5 | 26.24 | 0.011 | 0.007 | -0.009 | -0.247 | 0.031 | 0.041 | 0.052 | 0.342 |
| 6 | 27.84 | 0.009 | 0.004 | -0.008 | -0.232 | 0.031 | 0.040 | 0.053 | 0.346 |
| 7 | 31.40 | 0.008 | 0.001 | -0.010 | -0.230 | 0.036 | 0.046 | 0.061 | 0.365 |
| 8 | 51.26 | -0.004 | -0.017 | -0.015 | -0.094 | 0.156 | 0.159 | 0.127 | 0.115 |
| 9 | 52.28 | -0.004 | -0.009 | -0.004 | -0.033 | 0.169 | 0.131 | 0.076 | 0.039 |
| 10 | 53.86 | -0.001 | 0.002 | -0.001 | -0.002 | 0.095 | 0.025 | 0.015 | 0.012 |
| 11 | 54.94 | 0.002 | 0.000 | -0.000 | -0.000 | 0.023 | 0.011 | 0.008 | 0.003 |
| 12 | 56.66 | 0.002 | 0.000 | 0.000 | 0.000 | 0.010 | 0.004 | 0.002 | 0.000 |
| 13 | 57.30 | 0.001 | 0.000 | 0.000 | 0.000 | 0.009 | 0.003 | 0.001 | 0.000 |
| 14 | 58.00 | 0.001 | 0.000 | 0.000 | 0.000 | 0.008 | 0.002 | 0.001 | 0.000 |

**Table 2: Statistics for the comparison between RTTOV-gb and the line-by-line model R98 (Rosenkranz, 1998) with**
**the optimal RTTOV training configuration and the independent profile set. HATPRO channel number (CHAN #), the**
**channel central frequency, bias and RMS at elevation angles 90, 30, 19 and 10° are reported.**



### DEPENDENT PROFILE SET

| CHAN # | Frequency(GHz) | BIAS (K) 90° | 30° | 19° | 10° | RMS (K) 90° | 30° | 19° | 10° |
|---|---|---|---|---|---|---|---|---|---|
| 1 | 22.23 | -0.016 | 0.027 | -0.004 | -0.319 | 0.059 | 0.047 | 0.044 | 0.376 |
| 2 | 22.50 | -0.015 | 0.026 | -0.004 | -0.321 | 0.058 | 0.047 | 0.044 | 0.378 |
| 3 | 23.03 | -0.009 | 0.025 | -0.005 | -0.318 | 0.053 | 0.045 | 0.042 | 0.370 |
| 4 | 23.83 | 0.002 | 0.020 | -0.007 | -0.313 | 0.043 | 0.040 | 0.038 | 0.357 |
| 5 | 25.00 | 0.010 | 0.014 | -0.007 | -0.300 | 0.039 | 0.037 | 0.037 | 0.346 |
| 6 | 26.23 | 0.011 | 0.010 | -0.006 | -0.284 | 0.037 | 0.036 | 0.036 | 0.343 |
| 7 | 28.00 | 0.011 | 0.008 | -0.005 | -0.270 | 0.036 | 0.037 | 0.037 | 0.3500 |
| 8 | 30.00 | 0.011 | 0.006 | -0.004 | -0.266 | 0.038 | 0.040 | 0.040 | 0.366 |
| 9 | 51.25 | 0.024 | 0.018 | 0.013 | -0.088 | 0.177 | 0.171 | 0.128 | 0.104 |
| 10 | 51.76 | 0.024 | 0.019 | 0.013 | -0.056 | 0.189 | 0.164 | 0.111 | 0.066 |
| 11 | 52.28 | 0.025 | 0.019 | 0.012 | -0.029 | 0.203 | 0.149 | 0.085 | 0.034 |
| 12 | 52.80 | 0.029 | 0.020 | 0.008 | -0.011 | 0.207 | 0.116 | 0.052 | 0.014 |
| 13 | 53.37 | 0.019 | 0.017 | 0.002 | -0.002 | 0.181 | 0.068 | 0.022 | 0.005 |
| 14 | 53.85 | 0.012 | 0.007 | 0.001 | 0.000 | 0.120 | 0.029 | 0.010 | 0.003 |
| 15 | 54.40 | 0.006 | -0.000 | 0.001 | 0.000 | 0.055 | 0.012 | 0.006 | 0.002 |
| 16 | 54.94 | 0.004 | 0.001 | 0.000 | 0.000 | 0.023 | 0.007 | 0.003 | 0.001 |
| 17 | 55.50 | 0.002 | 0.001 | 0.000 | -0.000 | 0.013 | 0.004 | 0.002 | 0.000 |
| 18 | 56.02 | 0.001 | 0.000 | 0.000 | -0.000 | 0.009 | 0.003 | 0.001 | 0.000 |
| 19 | 56.66 | 0.001 | 0.000 | 0.000 | -0.000 | 0.007 | 0.002 | 0.000 | 0.000 |
| 20 | 57.29 | 0.001 | 0.000 | -0.000 | -0.000 | 0.005 | 0.001 | 0.000 | 0.000 |
| 21 | 57.96 | 0.001 | 0.000 | -0.000 | -0.000 | 0.004 | 0.001 | 0.000 | 0.000 |
| 22 | 58.80 | 0.000 | 0.000 | -0.000 | -0.000 | 0.004 | 0.001 | 0.000 | 0.000 |



### INDEPENDENT PROFILE SET

| CHAN # | Frequency(GHz) | BIAS (K) 90° | 30° | 19° | 10° | RMS (K) 90° | 30° | 19° | 10° |
|---|---|---|---|---|---|---|---|---|---|
| 1 | 22.23 | -0.008 | 0.022 | -0.003 | -0.284 | 0.049 | 0.046 | 0.042 | 0.157 |
| 2 | 22.50 | -0.008 | 0.021 | -0.005 | -0.279 | 0.048 | 0.046 | 0.043 | 0.158 |
| 3 | 23.03 | -0.002 | 0.020 | -0.006 | -0.277 | 0.042 | 0.045 | 0.042 | 0.154 |
| 4 | 23.83 | 0.007 | 0.017 | -0.008 | -0.274 | 0.035 | 0.044 | 0.045 | 0.164 |
| 5 | 25.00 | 0.012 | 0.011 | -0.009 | -0.263 | 0.033 | 0.043 | 0.051 | 0.206 |
| 6 | 26.23 | 0.011 | 0.007 | -0.009 | -0.247 | 0.031 | 0.041 | 0.052 | 0.236 |
| 7 | 28.00 | 0.010 | 0.004 | -0.008 | -0.232 | 0.031 | 0.040 | 0.053 | 0.257 |
| 8 | 30.00 | 0.008 | 0.002 | -0.009 | -0.228 | 0.033 | 0.043 | 0.057 | 0.273 |
| 9 | 51.25 | -0.005 | -0.018 | -0.016 | -0.094 | 0.156 | 0.160 | 0.128 | 0.067 |
| 10 | 51.76 | -0.005 | -0.014 | -0.010 | -0.061 | 0.162 | 0.149 | 0.105 | 0.039 |
| 11 | 52.28 | -0.005 | -0.009 | -0.004 | -0.037 | 0.170 | 0.131 | 0.077 | 0.020 |
| 12 | 52.80 | -0.004 | 0.000 | -0.001 | -0.015 | 0.169 | 0.098 | 0.044 | 0.015 |
| 13 | 53.37 | -0.003 | 0.007 | -0.003 | -0.005 | 0.145 | 0.056 | 0.021 | 0.015 |
| 14 | 53.85 | -0.002 | 0.002 | -0.002 | -0.002 | 0.097 | 0.026 | 0.015 | 0.012 |
| 15 | 54.40 | 0.000 | -0.002 | -0.001 | -0.001 | 0.047 | 0.015 | 0.011 | 0.007 |
| 16 | 54.94 | 0.002 | 0.000 | -0.000 | -0.000 | 0.023 | 0.011 | 0.008 | 0.003 |
| 17 | 55.50 | 0.002 | 0.000 | 0.000 | -0.000 | 0.016 | 0.007 | 0.005 | 0.001 |
| 18 | 56.02 | 0.002 | 0.000 | 0.000 | -0.000 | 0.013 | 0.005 | 0.003 | 0.001 |
| 19 | 56.66 | 0.001 | 0.000 | 0.000 | 0.000 | 0.010 | 0.004 | 0.002 | 0.000 |
| 20 | 57.29 | 0.000 | 0.000 | 0.000 | 0.000 | 0.009 | 0.003 | 0.001 | 0.000 |
| 21 | 57.96 | 0.000 | 0.000 | 0.000 | 0.000 | 0.008 | 0.002 | 0.000 | 0.000 |
| 22 | 58.80 | 0.000 | 0.000 | 0.000 | 0.000 | 0.007 | 0.002 | 0.000 | 0.000 |





**Table 3: Statistics for the comparison between RTTOV-gb and the line-by-line model R98 at MP-3000A channels with the optimal RTTOV training configuration, for both dependent (top) and independent (bottom) profile set. MP3000A channel number (CHAN #), the channel central frequency, bias and RMS at elevation angles 90, 30, 19 and 10° are reported.**

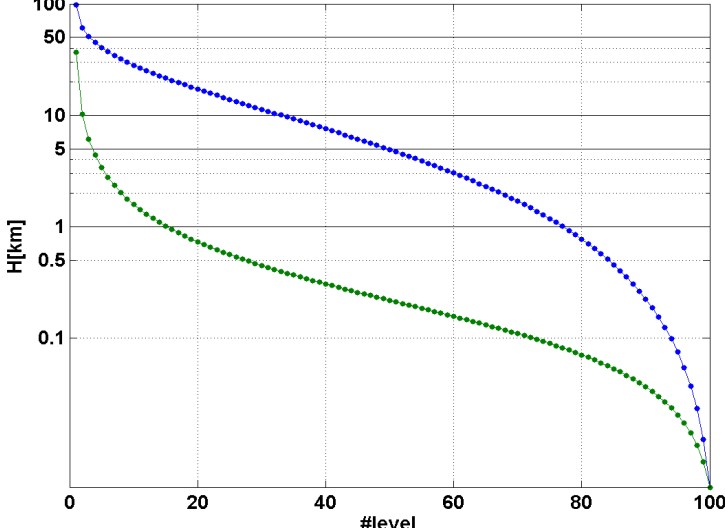

**Figure 1: Vertical spacing of profiles levels used for RTTOV in this analysis. Level altitudes and altitude differences between levels are reported respectively with blue and green lines. Note that y-axis is in logarithmic scale.**

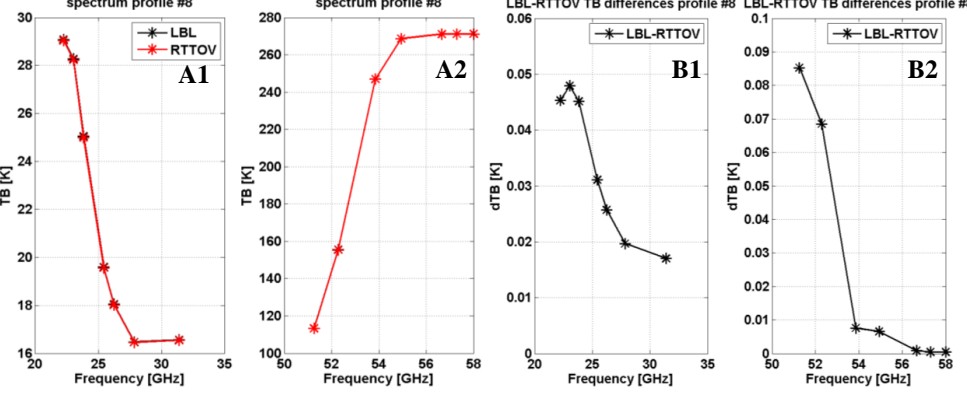

**Figure 2: (A1) TB at K-band channels (20-35 GHz) computed by RTTOV-gb (red stars) and LBL R98 (black stars) from profile #8 of the dependent set. (A2) Same as A1, but for V-band channels (50-60GHz). (B1) TB differences (LBL R98 minus RTTOV-gb) at K-band channels. (B2) Same as B1, but for V-band channels.**



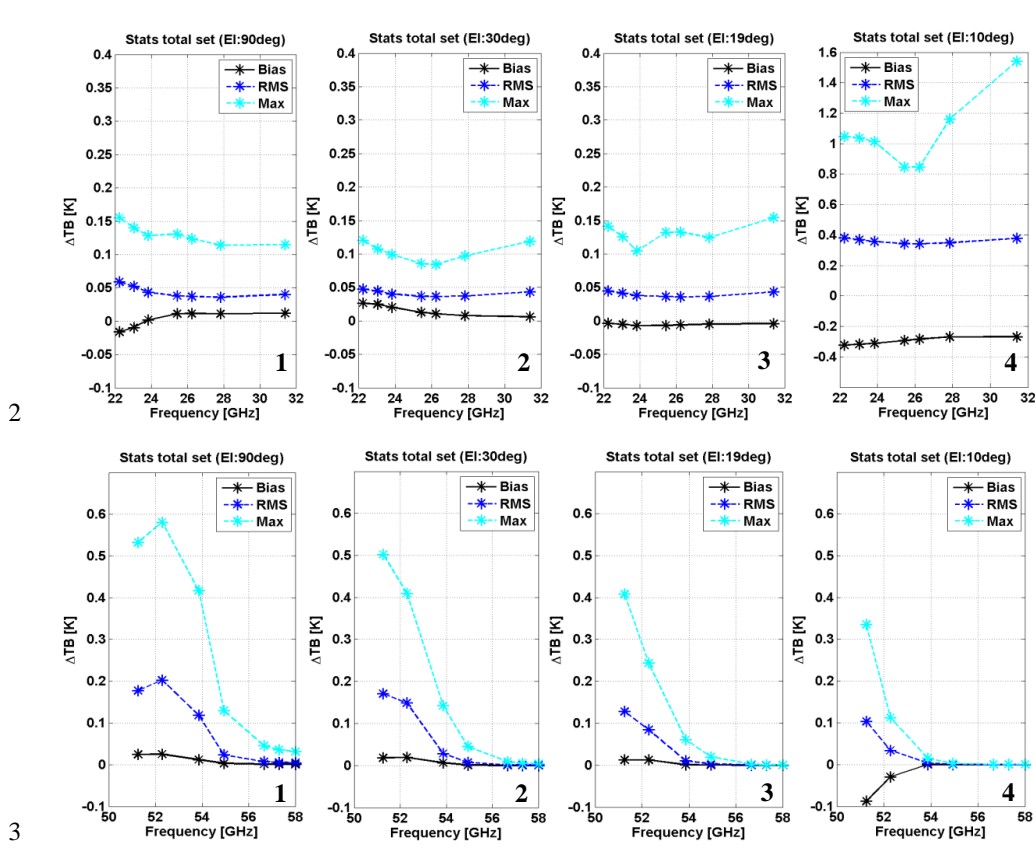

**Figure 3: Bias (black solid line), RMS (blue dashed line) and maximum (cyan dashed line) of TB difference between**
**RTTOV-gb and LBL R98 (Rosenkranz, 1998) for the dependent 83-profile set and the optimal training configuration.**
**Top panels:  K-band channels; Bottom panels: V-band channels. Panels number 1-2-3-4 report results at 90-30-19-10°**
**elevation angle, respectively. Note that top panel 4 has different y-axis scale with respect to the other top panels.**





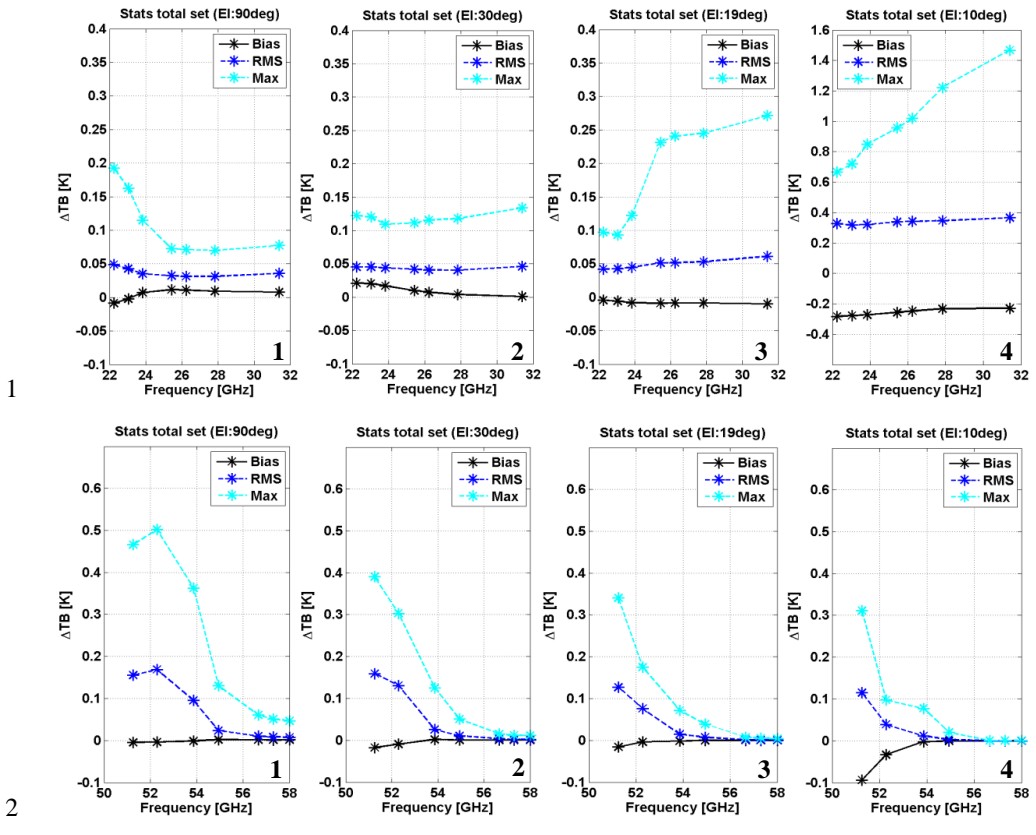

**Figure 4: Same as Figure 3 but for the independent 52-profile set. Top panels: K-band channels; Bottom panels: V-band channels. Panels number 1-2-3-4 report results at 90-30-19-10° elevation angle, respectively. Note that top panel 4 has different y-axis scale with respect to the other top panels.**





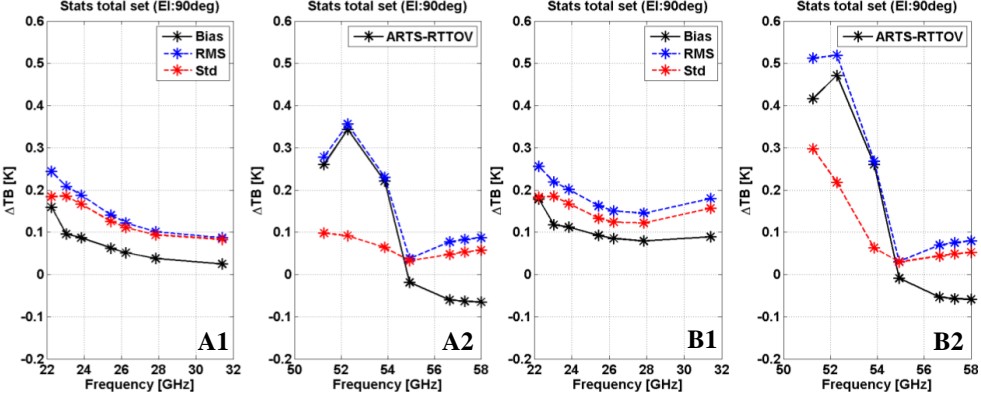

**Figure 5: Bias (black solid line), standard deviation (red dashed line) and RMS (blue dashed line) of TB differences between RTTOV-gb and the reference radiative transfer model ARTS (Eriksson et al., 2015), for both clear (A1-2) and cloudy (B1-2) sky conditions. Panels 1-2 are for K- and V-band channels, respectively. All panels report results at 90° elevation angle.**

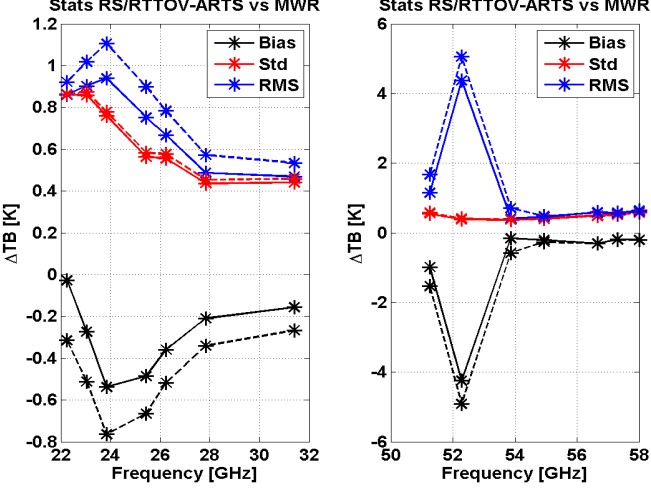

**Figure 6: Bias (black line), standard deviation (red line) and RMS (blue line) of differences between TB measured with MWR and TB simulated from radiosonde profiles respectively with RTTOV-gb (solid lines) and the reference radiative transfer model ARTS (dashed lines), both for clear-sky at 90° elevation angle.**





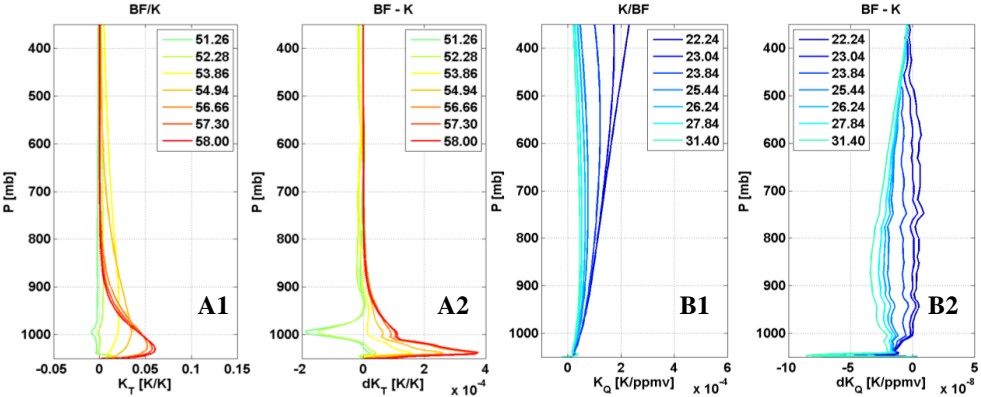

**Figure 7: Jacobians calculated with the RTTOV-gb BF method and K-module. Panels A1: temperature Jacobians for**
**V-band channels; Panels B1: absolute humidity for K-band channels. Note that BF method (solid) and K-module**
**(dashed) are not distinguishable as they nearly completely overlap. Panels A2 and B2 show Jacobian differences**
**between BF and K, respectively for temperature and absolute humidity.**

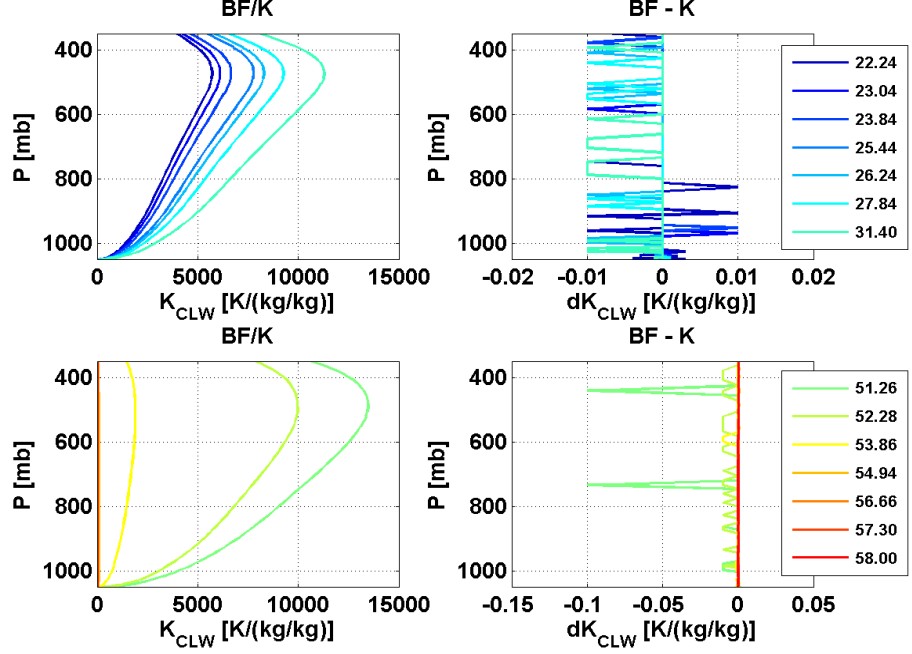

**Figure 8: Cloud Liquid Water Jacobians calculated with RTTOV-gb BF method and K-module (left) and Jacobian**
**differences between BF and K (right), respectively for K-band (top) and V-band (bottom) channels. Note that BF**
**method (solid) and K-module (dashed) are not distinguishable as they nearly completely overlap.**





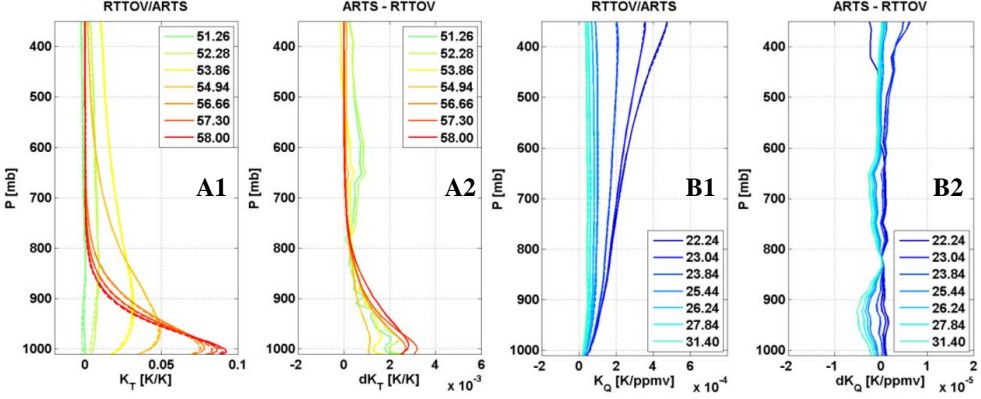

**Figure 9: As in Figure 7, but for Jacobians calculated with ARTS (solid line) and RTTOV-gb K-module (dashed line).**
**Panels A2 and B2 show Jacobian differences between ARTS and RTTOV-gb K-module, respectively for temperature**
**and absolute humidity.**

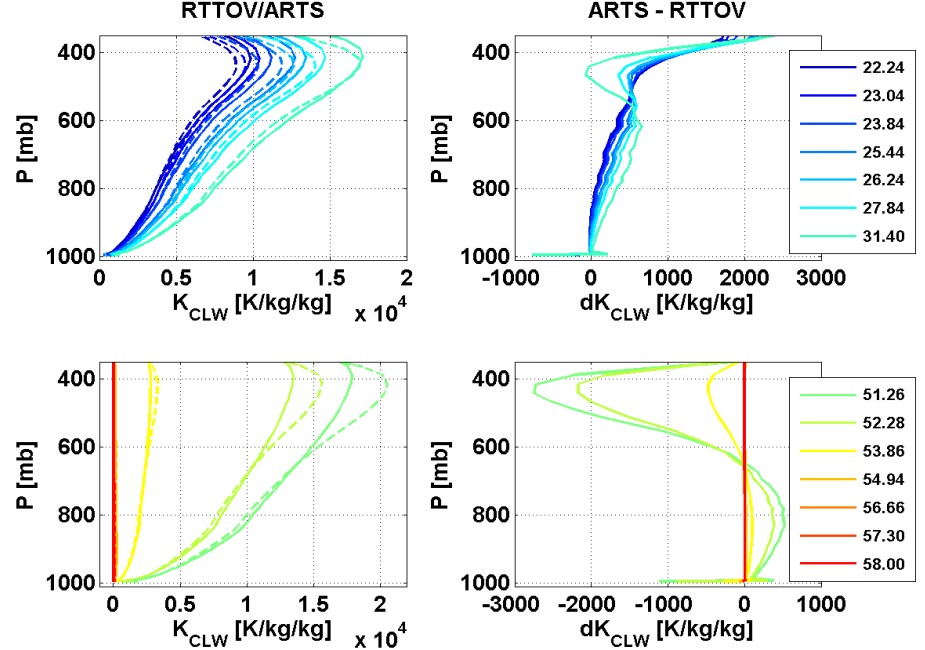

**Figure 10: As in Figure 8, but for Jacobians calculated with ARTS (solid line) and RTTOV-gb K-module (dashed**
**line).**





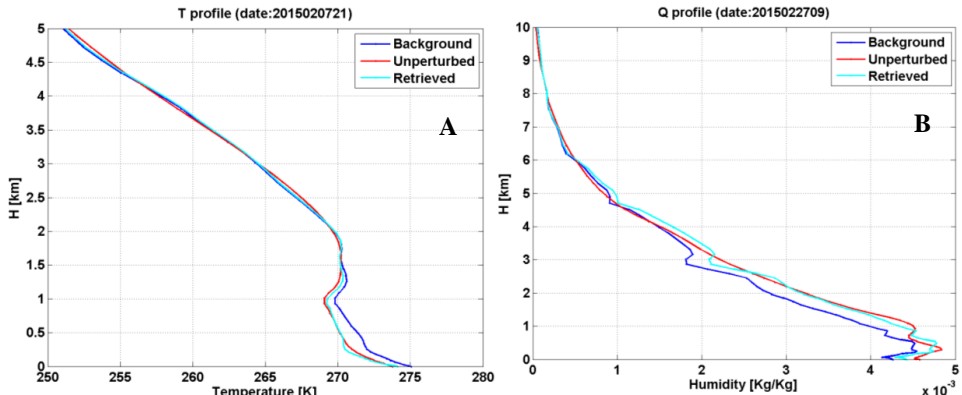

3   **Figure 11: Temperature (Panel A) and Humidity (Panel B) profiles of Background (blue line), Truth (red line) and**

4   **1D-Var retrievals (cyan line) for two clear-sky profiles.**

