# Peer review of "RTTOV-gb - Adapting the fast radiative transfer model"

_Geoscientific Model Development, 2016_

## Referee Comment (RC1) · Anonymous Referee #1 · 27 Jun 2016

The paper presents an adaptation of the RTTOV radiative transfer model for the ground based observation geometry, called RTTOV-gb. The regression coefficients are computed for a given set of predictors using the same training set that has been used to train RTTOV and the line-by-line absorption model of Rosenkranz 1998. RTTOV-gb has been validated against two radiative transfer models both using line-by-line absorption models and against real observations from a microwave radiometer where RTTOV-gb has been evaluated using colocated radiosonde profiles. The jacobians module of RTTOV-gb has also been validated against ARTS. Finally, RTTOV-gb has been used to perform 1D-var retrievals in a pure simulation environment (OSSE) and the authors

could show that the retrievals are closer to the truth than the background.

The paper is very well written, methodologically thorough and scientifically convincing. It is an excellent and very important contribution for the future assimilation of ground based microwave radiometers. It fits very well in the scope of GMD and I recommend to accept the paper for publication subject to some very minor revisions.

Specific comments

P4/l14: not clear if fixed pressure levels are used. I suggest to write "...are pressures of the fixed pressure levels..."

P5/l24: this is better referred to as "linear combination" than "polynomial".

P6/l10: say here which LBL model has been used.

P6/l10: is it justified to say that only 83 profiles cover the variability of humidity and temperature?

P6/l34: use "state vector" instead of "profile vector".

P7/l10: y should be bold in the denominator.

P8/l5: refer to table 2.

P8/l31: in this and the following sentence you use twice "underestimate" which does not seem to make sense. Please check.

P10/l8: why does it make sense to compare RTTOV-gb to two other implementations of R98? Would it not be more sensible to use different absorption models. For ARTS one could use individual LBL calculation rather than the predefined models. Please comment.

---

## Author Comment (AC1) · 30 Jun 2016

**REPLY TO REFEREE #1**

The authors are grateful to the referee for the positive feedback and the constructive comments. We accept them all and acknowledge the help in improving the manuscript. Our point by point replies are highlighted in red below as well as in the revised manuscript (to be submitted after the comments from Referee #2 will become available).

**Specific comments:**

**P4/l14: not clear if fixed pressure levels are used. I suggest to write "...are pressures of the fixed pressure levels. . ."**

Agreed. Text has been changed according to the referee comment in the revised manuscript.

**P5/l24: this is better referred to as "linear combination" than "polynomial".**

Agreed. Text has been changed according to the referee comment in the revised manuscript.

**P6/l10: say here which LBL model has been used.**

Agreed. Text concerning LBL model has been moved earlier in the revised manuscript, according to the referee comment.

**P6/l10: is it justified to say that only 83 profiles cover the variability of humidity and temperature?**

This profile set has been assembled at ECMWF to represent the range of variations in temperature and absorber amount found in the real atmosphere. This set was sampled from a large profile dataset containing 121,462,560 profiles generated using the experimental suite of the ECMWF forecasting system. To make this point clear, we added a new reference and the following sentence in the revised manuscript:

"It is important to emphasise that this profile set was carefully chosen from a set of more than 100 million profiles to represent a wide range of physically realistic atmospheric states (Matricardi, 2008)."

**P6/l34: use "state vector" instead of "profile vector".**

Agreed. Text has been changed according to the referee comment in the revised manuscript.

**P7/l10: y should be bold in the denominator.**

Thanks much for spotting this typo. It has been modified in the revised manuscript.

**P8/l5: refer to table 2.**

At P8/l5 we refer to Table 1, which presents the results for the dependent profile set. We refer to Table 2 at P9/l24, where the results for the independent profile set are discussed. Hope this clarifies, otherwise don't hesitate to let us know.

**P8/l31: in this and the following sentence you use twice "underestimate" which does not seem to make sense. Please check.**

**Agreed. We checked and confirm that "underestimate" is correct in both sentences. However, to make it more clear we modified the sentence as follows:**

**"However, in such a case where the optical depths are underestimated, then the atmosphere as a whole is too transparent".**

**P10/l8: why does it make sense to compare RTTOV-gb to two other implementations of R98? Would it not be more sensible to use different absorption models. For ARTS one could use individual LBL calculation rather than the predefined models. Please comment.**

**As stated in P9/l40 of the original manuscript, "In this analysis, ARTS settings for absorption model have been selected to adopt as much as possible the same absorption model as RTTOV-gb". Indeed, the goal of that analysis is to test the fast RT modeling (RTTOV-gb) with respect to accurate but slower LBL calculation, all the other settings (including gas absorption model) being equal. To make this point more clear the following sentence has been added:**

**"Since the goal of this analysis is to test the fast RT modeling (RTTOV-gb) with respect to accurate LBL calculation, all other settings being equal, ARTS settings for absorption model have been selected to adopt as much as possible the same absorption model as RTTOV-gb".**

---

## Referee Comment (RC2) · Anonymous Referee #2 · 4 Jul 2016

Numerical weather prediction (NWP) uses input from a large number of observation systems, but ground-based microwave radiometers (MWRs) are not yet considered despite that these instrument should provide valuable additional information. The preferred manner to incorporate remote sensing data is to assimilate measured radiances (instead of making use of external retrievals) and this requires that a radiative transfer model for the observation system is at hand. The forward models used at NWP centres presently lack treatment of MWRs, and this manuscript presents an extension of the RTTOV model to remove this shortcoming. This is a valuable contribution that fits with the scope of the journal.

That said, the value of the manuscript depends critically on if the extension actually becomes part of future releases of RTTOV, or not. The manuscript is not clear on this point, the extension is just said to be "under discussion". My review below will be based on the assumption that the extension will be accepted. If this will not happen, I think the manuscript has to change focus. This is the case as the manuscript is of little interest for extending any other fast forward model, the discussion is quite specific for RTTOV. Further, the 1D-var examples shown (Sec 3.4) could have been performed with many other forward models, i.e. they could have been obtained without extending RTTOV. That is, I encourage the authors to confirm that the extension will be incorporated into the official RTTOV version.

Specific comments:

Page2, line 8: Why parenthesis around "in cloudy areas"? It causes confusion, is it from WMO, or your own comment?

Page 2, line 9: How do you define the acronym? It is below used to mean both one or several radiometers.

Page 3, line 33: What are the approximations? Make sure that what you write here is synced with the text in page 4, line 7.

Eq 2: It is highly confusing to use tau as symbol for transmittance. Tau is the standard symbol for optical depth (as you also noticed on page 4, line 34). Please consider to change notation. Maybe you follow the standard notation in RTTOV, but I don't see that as a sufficiently strong argument.

Page 3, line 37: To define a transmittance, you must give two points (here just TOA).

Page 4, line 22: The term describes the contribution of cosmic background radiation.

Page 5, line 8: For consistency, unit for CLW should be added.

Page 5, line 23: Don't see the logic behind "Thus" here. Is there not even a contradiction between the sentence before and including "Thus"? Just remove the first sentence and "Thus" in the second one? Anyhow, I did not follow this part before reaching Eq 9.

Page 6, line 13: By using the word "optimally" you imply that you have considered all possible combinations. Is that really true?

Page 7, line 15: What is the transpose of software code?

Page 7, line 26: Should it be "employment"?

Sec 3.1: A lot of details (with low interest) in this section. Particularly, I don't see the point with all the tables. I don't argue about this further, just encourages the authors to consider a shortening of the section.

Page 8, line 13: Here you use V-band without introduction. Very few knows the definition of these "bands". Why not just use frequencies?

Comparison with ARTS: According to Fig 5 there is a significant deviation to ARTS already for "clear sky", but my understanding of the text is that the same absorption models are used for oxygen and water absorption. So why is the "bias" bigger here than in Fig 3? This seems to indicate a deviation already between LBL R98 and ARTS. Have you checked this?

Sec 3.4: When reading about OSSE in the abstract, I expected much more than this. The section shows some test retrievals (by 1d-var/OEM) with a priori taken from a model. I don't know if there exists an official definition of OSSE, but I expected to find a much more elaborated exercise (e.g. 4d-var) when OSSE was mentioned. That is, I recommend to not denote this as an OSSE.

Qpack + ARTS speed: As I did not understand the relation between Qpack2 and ARTS (that should be explained clearer in the manuscript), I asked a colleague that I had heard mention Qpack. He happened to notice the comment on calculation speed and then said that the speed of ARTS depends critically on several settings. Moreover, he also said that using ARTS through Qpack2 means a lot of overhead and if the

Jacobian calculation is clocked inside Qpack2 this does not necessarily reflect the true performance of ARTS. Without understanding the details here, I must still ask if these issues have been considered?

In addition, is the calculation speed of RTTOV-gb really the main contribution? Considering the relatively low number of MWRs and their relatively low number of channels, compared to all satellite data, I assume that NWP centre easily could afford a slow forward model to assimilate MWR data. However, they would still tend to ignore MWR data as long as those data require usage of an additional forward model. This another way to express why it is critical if this extension will be part of the official RTTOV or not.

Sec 6: Place this section before Conclusions.

---

## Author Comment (AC2) · 14 Jul 2016

**REPLY TO REFEREE #2**

The authors are grateful to the referee for the constructive comments. We revised the manuscript accordingly. Our point by point replies are shown in red hereafter, while modifications to the text are highlighted in yellow within the revised manuscript (modifications in red are made according to the referee #1 comments).

**General comment:**

**Numerical weather prediction (NWP) uses input from a large number of observation systems, but ground-based microwave radiometers (MWRs) are not yet considered despite that these instrument should provide valuable additional information. The preferred manner to incorporate remote sensing data is to assimilate measured radiances (instead of making use of external retrievals) and this requires that a radiative transfer model for the observation system is at hand. The forward models used at NWP centres presently lack treatment of MWRs, and this manuscript presents an extension of the RTTOV model to remove this shortcoming. This is a valuable contribution that fits with the scope of the journal.**

**That said, the value of the manuscript depends critically on if the extension actually becomes part of future releases of RTTOV, or not. The manuscript is not clear on this point, the extension is just said to be "under discussion". My review below will be based on the assumption that the extension will be accepted. If this will not happen, I think the manuscript has to change focus. This is the case as the manuscript is of little interest for extending any other fast forward model, the discussion is quite specific for RTTOV.**

**Further, the 1D-var examples shown (Sec 3.4) could have been performed with many other forward models, i.e. they could have been obtained without extending RTTOV. That is, I encourage the authors to confirm that the extension will be incorporated into the official RTTOV version.**

We agree with the referee this aspect is important. However, it does not depend totally on the authors. Indeed, for RTTOV-gb to become part of future releases of RTTOV it would require additional efforts from NWP SAF for documentation, maintenance, user support, etcetera. This is currently not planned within NWP SAF activities. However, we started the process and this issue has been recently discussed at the NWP SAF Steering Group meeting. We are also looking for opportunities for funding these activities.

The alternative option is to release RTTOV-gb as a stand-alone package through the TOPROF website. Users will still have to sign a license agreement with NWP SAF before they can download the package, but TOPROF will be responsible for documentation, maintenance, user support, etcetera.

If the NWP SAF Steering Group will not recommend the integration of RTTOV-gb in future RTTOV releases, the alternative option will definitely be pursued. This may be sought as an intermediate step, which would minimize the NWP SAF efforts without hampering the development of ground-based MWR data assimilation. Based on the successful outcome of these activities, the integration of RTTOV-gb in RTTOV may be reconsidered in the future.

In this perspective, RTTOV-gb will be made available one way or the other and thus we believe this issue shall not affect the value of the manuscript.

To make this point clearer, we modified the text as follows in the revised manuscript: "The conditions of release of RTTOV-gb are currently under discussion among NWPSAF and COST action TOPROF. This may happen through an integration of RTTOV-gb into future RTTOV releases or as a stand-alone package disseminated through the TOPROF website."

**Specific comments:**

**Page 2, line 8: Why parenthesis around "in cloudy areas"? It causes confusion, is it from WMO, or your own comment?**

The statement "in cloudy areas" comes from WMO. We removed parenthesis in the revised manuscript.

**Page 2, line 9: How do you define the acronym? It is below used to mean both one or several radiometers.**

The acronym "MWR" stands for "microwave radiometers" (i.e. plural). We modified the revised manuscript accordingly.

**Page 3, line 33: What are the approximations? Make sure that what you write here is synced with the text in page 4, line 7.**

The Radiative Transfer Equation in equation (2) assumes negligible scattering. This is stated just after Equation (2). Note that the liquid water contribution is included in the transmittance profile just as other absorbing gaseous species. This is consistent with what stated at page 4, line 7.

**Eq 2: It is highly confusing to use tau as symbol for transmittance. Tau is the standard symbol for optical depth (as you also noticed on page 4, line 34). Please consider to change notation. Maybe you follow the standard notation in RTTOV, but I don't see that as a sufficiently strong argument.**

We agree with the referee that tau is a common notation for optical depth. We used tau for transmittances in the manuscript to be consistent with the RTTOV nomenclature (used in several papers and ECMWF technical memoranda) and with the names of the variables used in the RTTOV/RTTOV-gb code. The authors here aim to be as consistent as possible with the RTTOV notation in public literature and code. This is also strategic in the perspective of RTTOV-gb integration into RTTOV official releases (as already discussed in the general comment above). We hope this makes a sufficiently strong argument.

**Page 3, line 37: To define a transmittance, you must give two points (here just TOA).**

Agreed. Text has been changed according to the referee comment in the revised manuscript.

**Page 4, line 22: The term describes the contribution of cosmic background radiation.**

Agreed. Text has been changed according to the referee comment in the revised manuscript.

**Page 5, line 8: For consistency, unit for CLW should be added.**

**Agreed. The CLW unit has been added in the revised manuscript.**

**Page 5, line 23: Don't see the logic behind "Thus" here. Is there not even a contradiction between the sentence before and including "Thus"? Just remove the first sentence and "Thus" in the second one? Anyhow, I did not follow this part before reaching Eq 9.**

**Agreed. The sentence has been modified according to the referee comment in the revised manuscript.**

**Page 6, line 13: By using the word "optimally" you imply that you have considered all possible combinations. Is that really true?**

**The statement "optimally" refers to the best set of elevation angles found among the considered configurations. For each configuration, we chose 6 elevation angles within the elevation angle range of commercial MWR (90-0°). Note that we did not consider very low elevation angles, since the predictors were originally developed for satellite simulations up to 15° elevation angle.**

**However, we agree with the reviewer that the statement "optimally" may be misleading and thus decided to remove it in the revised manuscript.**

**Page 7, line 15: What is the transpose of software code?**

**As opposite to the TL code, which inputs state vector profile and outputs radiances, the AD code inputs radiances and outputs state vector profiles. Thus, the AD code is often referred to as the transpose of the TL code. But we agree with the referee this sounds like a jargon and thus we removed it in the revised manuscript.**

**Page 7, line 26: Should it be "employment"?**

**Sorry, we meant "exploitation". Thanks for spotting this typo. It has been modified in the revised manuscript.**

**Sec 3.1: A lot of details (with low interest) in this section. Particularly, I don't see the point with all the tables. I don't argue about this further, just encourages the authors to consider a shortening of the section.**

**In order to address the referee's comment, we shortened Section 3.1 in the revised manuscript. However, we prefer to keep the Tables, as we believe they are useful to the discussion and provide a reference for potential future RTTOV-gb users.**

**Page 8, line 13: Here you use V-band without introduction. Very few knows the definition of these "bands". Why not just use frequencies?**

Agreed. We now introduce the frequency bands in the revised manuscript: "Channels from 22 to 31 GHz are in the so called K-band while channels from 51 to 58 GHz are in the so-called V-band." We prefer to use "V-band" and "K-band" instead of repeating the channel frequencies every time.

**Comparison with ARTS: According to Fig 5 there is a significant deviation to ARTS already for "clear sky", but my understanding of the text is that the same absorption models are used for oxygen and water absorption. So why is the "bias" bigger here than in Fig 3? This seems to indicate a deviation already between LBL R98 and ARTS. Have you checked this?**

We do expect differences larger in Fig. 5 (RTTOV-gb vs ARTS) than in Fig. 3 (RTTOV-gb vs LBL), due to the reasons discussed in the manuscript: (1) larger regression error (Fig. 5 shows results from a dataset completely independent from the training set) and (2) larger computing differences (ARTS uses radiative transfer routines completely independent from the ones used by RTTOV-gb and LBL).

However, we agree with the reviewer that Fig.5 shows unexpected large bias at the three most transparent V-band channels (i.e. 51.26, 52.28, and 53.25 GHz). Indeed, the difference between LBL R98 and ARTS confirms systematic differences below 0.1 K at all but the above three channels. This may be due to small differences in the implementation of the R98 gas absorption and/or the radiative transfer code. We discussed this with ARTS developers and plans were made to further investigate. We added the following text to Section 3.1 for clearing this point: "This is dominated by a bias contribution induced by systematic differences found between LBL and ARTS at these three channels (~0.3-0.5 K, not shown). This may be caused by small differences in the implementation of the R98 gas absorption and/or the radiative transfer code. This issue is currently under investigation, though its understanding goes beyond the scope of this paper."

**Sec 3.4: When reading about OSSE in the abstract, I expected much more than this. The section shows some test retrievals (by 1d-var/OEM) with a priori taken from a model. I don't know if there exists an official definition of OSSE, but I expected to find a much more elaborated exercise (e.g. 4d-var) when OSSE was mentioned. That is, I recommend to not denote this as an OSSE.**

As specified in the abstract, the acronym "OSSE" stands for "Observing-System Simulation Experiment". We neither know of an official definition. To our understanding (*), OSSE requires simulated observations with simulated errors to be drawn from a simulated atmosphere and provided to a data assimilation system to produce estimates of the atmospheric state. An OSSE involves the following steps: 1) Generate a "nature" atmosphere; 2) Compute synthetic observations; 3) Assimilate the synthetic observations; and 4) Assess the impact. These steps have been followed in our analysis. The only difference with respect to the elaborated exercise the referee mention to is that we perform a 1D- instead of a 3D- or 4D-VAR experiment. But this is clearly stated in the abstract ("RTTOV-gb has been applied as the forward model operator within a 1-Dimensional Variational (1D-Var) software tool in an Observing-System Simulation Experiment"). Therefore, we prefer to leave it as is. A more extensive evaluation of RTTOV-gb in a 3D-Var experiment is planned as subject of our future work.
(*) e.g. see

https://cimss.ssec.wisc.edu/model/osse/osse.html
http://gmao.gsfc.nasa.gov/projects/osse/

**Qpack + ARTS speed: As I did not understand the relation between Qpack2 and ARTS (that should be explained clearer in the manuscript), I asked a colleague that I had heard mention Qpack. He happened to notice the comment on calculation speed and then said that the speed of ARTS depends critically on several settings. Moreover, he also said that using ARTS through Qpack2 means a lot of overhead and if the Jacobian calculation is clocked inside Qpack2 this does not necessarily reflect the true performance of ARTS. Without understanding the details here, I must still ask if these issues have been considered?**

In the revised manuscript, we chose to remove the mention to Qpack2. Qpack2 is really not relevant here, as Jacobians can be computed directly from ARTS. Indeed, tests on computing speed were made directly on ARTS (i.e. no use of Qpack2).

Nonetheless, we agree with the Referee that ARTS computing speed depends on several settings. We discussed with the ARTS developers a way to perform this comparison properly, and realized it needs a close collaboration of the two teams. We leave this for future work, as it goes beyond the scope of the present manuscript.

To satisfy the Referee comment, we keep just a conservative statement in the revised manuscript: "Of course the priority of LBL models is more accuracy than speed, though settings may be tuned to improve the computation performances. Although a detailed analysis on computation speed goes beyond the scope of this paper, we found that RTTOV-gb is faster than our implementation of ARTS (Martinet et al. 2015) for both the direct and Jacobian calculations."

Similarly, in the Summary: "As expected, RTTOV-gb demonstrates to be faster than the line-by-line models such as ARTS for both the direct and the Jacobians calculation"

**In addition, is the calculation speed of RTTOV-gb really the main contribution? Considering the relatively low number of MWRs and their relatively low number of channels, compared to all satellite data, I assume that NWP centre easily could afford a slow forward model to assimilate MWR data. However, they would still tend to ignore MWR data as long as those data require usage of an additional forward model. This another way to express why it is critical if this extension will be part of the official RTTOV or not.**

We agree with the referee that the calculation speed of RTTOV-gb is not the main contribution, due to the relative low number of ground-based MWR observations. We concur that NWP centers hardly accept new forward models, as this means additional efforts in terms of implementation and maintenance. This is the reason why we decided to adapt RTTOV despite the availability of other RT codes, whose speed may have been enough for the purpose.

We concur the main point is that RTTOV-gb works exactly the same as RTTOV from the user perspective, making the technical overheads for implementation and maintenance minimal. This shall make the road to ground-based MWR data assimilation at NWP centers easier.

Following the referee's comment, we include the following text to the revised manuscript (Section 4): "As from the user perspective RTTOV-gb works exactly the same as RTTOV, its implementation and

maintenance shall require minimal technical overheads at those NWP centers already using RTTOV. This shall facilitate the road towards the data assimilation of ground-based MWR worldwide."

**Sec 6: Place this section before Conclusions.**

We moved Section 6 (Code and data availability) one section earlier, just after Section 4 (Summary). This follows the GMD "Manuscript types" guidelines: "Inclusion of Code and/or data availability sections is mandatory for all papers and should be located at the end of the article, after the conclusions, and before any appendices or acknowledgements."

See the link below for more details:

http://www.geoscientific-model-development.net/about/manuscript_types.html

---

## Author Comment (AC3) · 14 Jul 2016

**REPLY TO REFEREE #1 – Add on**

After reading the comments from Referee #2 we modified to shorten Section 3.1. Thus, one of our reply to Referee #1 is changed slightly as follows.

**Specific comments:**

**P8/l31: in this and the following sentence you use twice "underestimate" which does not seem to make sense. Please check.**

Agreed. We checked and confirm that "underestimate" is correct in both sentences. However, to make it more clear we modified the text as follows:

"For the satellite (down-looking) case, these effects tend to compensate due to a warmer background (e.g. overestimated optical depths cause more emission from the atmosphere but less contribution from the relative warmer background)".

---

## Author Response (AR2)

REPLY TO THE TOPICAL EDITOR COMMENTS

Dear Dr. K. Gierens,

We would like to thank you for your careful review. We accepted all the changes you requested, and highlighted the new text in the revised manuscript below.

Please, find below the point-by-point reply to your comments.

Best regards.

**Page 2, 2nd line from bottom: "dimensional"**

**Thanks for spotting this typo. We have now corrected the text in the revised manuscript.**

**Equation 9: I suggest to write "X_ijk(T,q,theta)" instead of simply X_ijk, in order to let the reader see the dependencies.**

**We added the dependences of X_jk in Equation 9 in the revised manuscript.**

**Page 10, line 3: replace "larger statistics" by "larger differences" or something like this. I don't know what you mean with "larger statistics".**

**We changed the text in the revised manuscript.**

[revised manuscript text omitted]